# `SR-CACO-2`: A Dataset for Confocal Fluorescence Microscopy Image Super-Resolution

**Soufiane Belharbi**[1] **Mara KM Whitford**[2,3] **Phuong Hoang**[2] **Shakeeb Murtaza**[1]
**Luke McCaffrey**[2,3,4] **Eric Granger**[1]

[1] LIVIA, ILLS, Dept. of Systems Engineering, ETS Montreal, Canada
[2] Goodman Cancer Institute, McGill University, Montreal, Canada
[3] Dept. of Biochemistry, McGill University, Montreal, Canada
[4] Gerald Bronfman Dept. of Oncology, McGill University, Montreal, Canada
{soufiane.belharbi, eric.granger}@etsmtl.ca,
{mara.whitford, phuong.hoang2}@mail.mcgill.ca,
shakeeb.murtaza.1@ens.etsmtl.ca, luke.mccaffrey@mcgill.ca

## Abstract

Confocal fluorescence microscopy is one of the most accessible and widely used imaging techniques for the study of biological processes at the cellular and subcellular levels. Scanning confocal microscopy allows the capture of high-quality images from thick three-dimensional (3D) samples, yet suffers from well-known limitations such as photobleaching and phototoxicity of specimens caused by intense light exposure, which limits its use in some applications, especially for living cells. Cellular damage can be alleviated by changing imaging parameters to reduce light exposure, often at the expense of image quality. Machine/deep learning methods for single-image super-resolution (SISR) can be applied to restore image quality by upscaling lower-resolution (LR) images to produce high-resolution images (HR). These SISR methods have been successfully applied to photo-realistic images due partly to the abundance of publicly available datasets. In contrast, the lack of publicly available data partly limits their application and success in scanning confocal microscopy. In this paper, we introduce a large scanning confocal microscopy dataset named `SR-CACO-2` that is comprised of low- and high-resolution image pairs marked for three different fluorescent markers. It allows to evaluate the performance of SISR methods on three different upscaling levels (`X2`, `X4`, `X8`). `SR-CACO-2` contains the human epithelial cell line Caco-2 (ATCC HTB-37), and it is composed of 2,200 unique images, captured with four resolutions and three markers, that have been translated in the form of 9,937 patches for experiments with SISR methods. Given the new `SR-CACO-2` dataset, we also provide benchmarking results for 16 state-of-the-art methods that are representative of the main SISR families. Results show that these methods have limited success in producing high-resolution textures, indicating that `SR-CACO-2` represents a challenging problem. The dataset is released under a Creative Commons license (CC BY-NC-SA 4.0), and it can be accessed freely. Our dataset, code and pretrained weights for SISR methods are publicly available: https://github.com/sbelharbi/sr-caco-2.

## 1 Introduction

Confocal fluorescence microscopy is a standard imaging technique in biological and biomedical research [32, 33, 76]. It can resolve cell and tissue structures at a maximum lateral resolution of about 200nm, providing high-quality images at a moderate cost [79]. Point scanning confocal microscopy and spinning disk confocal microscopy are two advanced imaging techniques used for

38th Conference on Neural Information Processing Systems (NeurIPS 2024) Track on Datasets and Benchmarks.

high-resolution fluorescence imaging. Point scanning confocal microscopy uses a single laser beam to sequentially scan each point of the specimen, providing high-resolution images with excellent optical sectioning capability. This method is ideal for the detailed examination of thick three-dimensional (3D) specimens and allows for precise control over imaging parameters. However, it can be relatively slow and may cause photobleaching due to prolonged exposure to the laser [18, 36]. In contrast, spinning disk confocal microscopy uses a disk with multiple pinholes to simultaneously scan multiple points of the specimen. This results in much faster image acquisition, making it suitable for live cell imaging and dynamic processes. Spinning disk confocal microscopy also reduces photobleaching and phototoxicity compared to point scanning. However, its optical sectioning capability is generally lower, and it does not provide the same level of detail for thick 3D specimens. Therefore, an ongoing challenge with confocal microscopy of thick 3D samples is optimizing imaging parameters (laser intensity, exposure duration, scanning speed) to obtain high-quality images while minimizing phototoxicity. This is particularly challenging when capturing large 3D image stacks or time-lapse videos, where samples are repeatedly imaged 10s or 100s of times.

The success of microscopy is however limited due to the photobleaching of fluorescent probes caused by the excitation light of the emitted laser. Additionally, long exposure to this light can also cause phototoxicity of cells which leads to their damage and even death, further limiting the usage of such techniques in live cell imaging [18, 36]. Moreover, high-resolution images require long exposure to light which increases the risk of these issues and impedes the observation of instantaneous inter-cellular events. Different approaches have been considered to limit these issues, such as pulsed excitation [61], specialized culture media [6, 7], or more sophisticated techniques such as controlled light exposure microscopy [10, 34]. However, these techniques remain cumbersome, expensive, and less general. Another common and practical approach considers imaging with fluorescence microscopy with short exposure time or low excitation light intensity. This can lead to low-quality images which are subsequently improved using image-enhancing techniques [3, 9, 74, 75, 84].

Deep learning models have recently provided significant improvements in diverse image analysis tasks [26] and biosciences [43, 69]. In particular, they have allowed for considerable advances in single-image super-resolution (SISR) [92] where high-resolution (HR) images are restored from low-resolution (LR) photo-realistic images [52, 85, 95, 99]. However, the success of these models relies heavily on the availability of large-scale public datasets to train a deep SISR model. Few works in microscopy imaging aim to leverage SISR [8, 15, 35, 51, 60, 63, 65]. This extends to different microscopy techniques [79] including standard fluorescence microscopy such as confocal [76], Structured Illumination Microscopy (SIM) [27], STimulated Emission Depletion (STED) microscopy [31], Single-Molecule Localization Microscopy (SMLM) such as PhotoActivated Localization Microscopy (PALM) [5], and Stochastic Optical Reconstruction (STORM) [66]. Some of these works enhance image quality [8, 35, 60, 63], while others aim to upscale images by a factor [51, 65]. In contrast with standard SISR evaluations on photo-realistic images, where an LR image is typically a synthetic downsampled version of an HR image, SISR models for microscopy images deal with real LR images. This translates into a difficult task since real LR images are produced through a different unknown process than a deterministic interpolation which is relatively easy to learn. Despite the success of these SISR models, the availability of datasets remains an issue in microscopy, where datasets are private and inaccessible. This data unavailability hinders progress in microscopy imaging research and prevents leveraging the potential of deep SISR models.

Our work addresses the lack of public datasets for SISR in fluorescence microscopy. To the best of our knowledge, no public dataset exists for this modality. SR-CACO-2 is a new confocal fluorescence dataset proposed for bio-imaging with pairs of HR and (real) LR images for SISR. It is based on human epithelial cell line Caco-2 [46] (ATCC HTB-37). The imaged tiles (see Fig. 2) capture epithelial cells isolated from colon tissue with colorectal adenocarcinoma. Three different proteins are marked and imaged, yielding different views of a cell: Survivin (CELL0), E-cadherin or Tubulin (CELL1), and Histone H2B (CELL2). The dataset is composed of 2,200 unique images forming 22 tiles with the HR tiles measuring $9,300 \times 9,300$ pixels. They are captured by laser scanning over a regular raster. Three different LR scales are provided (Fig. 1): /2, /4, and /8 in addition to the HR scale. The SR-CACO-2 dataset is designed for convenient evaluation of machine learning models – patches are cropped to a size $512 \times 512$ from HR tiles, and their corresponding patches from all LR tiles at different scales. These patches are only comprised of regions of interest (ROI) (*i.e.*, cells), while irrelevant regions with black backgrounds are discarded. This allows for the collection of $9,937$

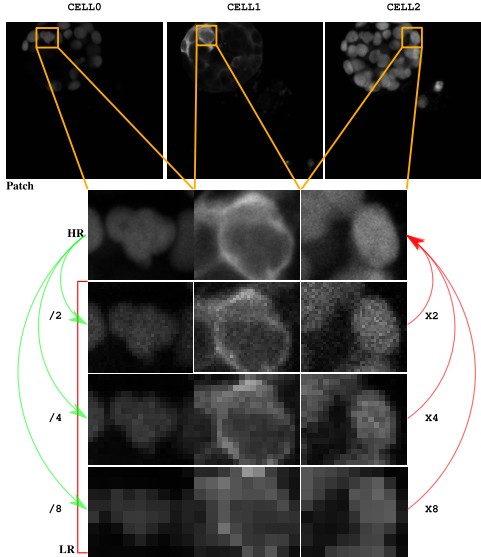

Figure 1: Illustration of SR-CACO-2 patch content for cells CELL0, CELL1, and CELL2, and for the HR patch and its corresponding three LR patches (/2, /4, /8). The HR patch size is $512 \times 512$, while the LR patch sizes are $256 \times 256$, $128 \times 128$, and $64 \times 64$ for scales /2, /4, and /8, respectively.

patches, per scale and cell type, that can be directly employed for the training and testing of deep SISR models.

**Our main contributions are summarized as follows.**
**(1)** A large, diverse, and challenging dataset, SR-CACO-2, for confocal fluorescence SISR microscopy is proposed. It captures an epithelial cell line derived from colorectal adenocarcinoma grown as 3D spheroids. SR-CACO-2 contains 2,200 unique images in HR format and three corresponding real LR versions /2, /4, and /8 covering three separate protein markers: Survivin, E-cadherin or Tubulin, and Histone H2B.
**(2)** A reproducible experimental protocol is introduced to perform machine learning experiments. It allows preparing the dataset for experiments with SISR methods.
**(3)** An extensive benchmarking study on 16 representative SISR methods is provided to assess the quality of super-resolved images. These results show that state-of-the-art SISR methods yield smooth images and fail to correctly produce accurate HR textures across all scales. The full SR-CACO-2 dataset (tiles and patches), code, and pretrained weights of methods are made public. The dataset is freely available under a Creative Commons license (CC BY-NC-SA 4.0).
**(4)** Images produced by all SISR methods are analyzed to assess their efficiency in downstream biology tasks for cell object (nucleus) detection and segmentation. Although they provide poor visual quality, several methods achieve promising results on these tasks compared to LR and HR.

## 2 The SR-CACO-2 Dataset

SR-CACO-2 is a dataset suitable for designing and evaluating machine/deep learning methods that can perform SISR in fluorescence microscopy imaging. In particular, image tiles of size $\sim 9k \times 9k$ are comprised of $10 \times 10$ unique images, and capture the human epithelial cell line Caco-2. These cells are a well-established model for studying mitotic spindle orientation and epithelial cell polarity. As they are cell lines, they can be cultured over long periods without the requirement for repeated re-isolation from tissue. They are also easily modified with lentivectors to overexpress or knock-down proteins of interest, or to introduce fluorescently-tagged proteins for use in live imaging.

The dataset was captured via fixed-cell imaging since it prevents the cells from moving, allowing for accurate capturing of all scales. The captured images allow training of SISR models that can potentially be used for live-imaging videos. Three proteins involved in cell division were used, as cell division is a behavior commonly studied via live imaging. First, **mCherry-Histone H2B** (CELL2, bright): Histone H2B marks chromatin (the DNA inside cells). It is tagged with the red

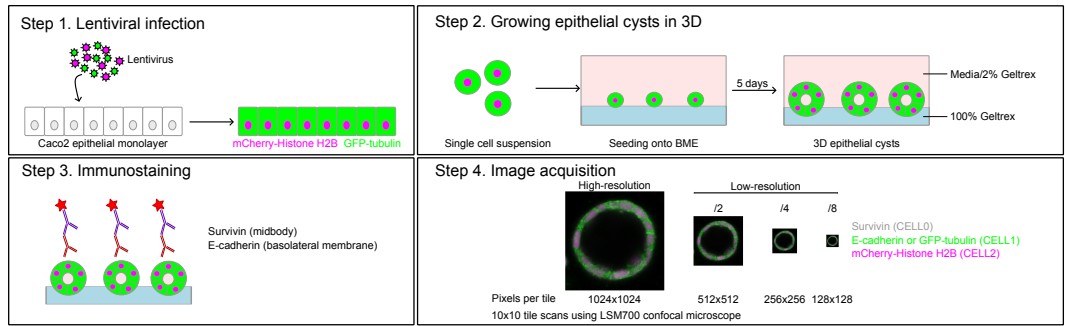

Figure 2: Methodology for capturing the `SR-CACO-2` dataset follows these steps:

**Step 1**: Lentiviral infection. mCherry-Histone H2B and GFP-tubulin lentivirus are added to a monolayer of Caco-2 epithelial cells. Infection of the monolayer with the viruses results in permanent modification of the cells to expression mCherry-Histone H2B (Magenta circles = red fluorescent chromosomes) and GFP-tubulin (Green rectangles = green fluorescent microtubules).

**Step 2**: Growing 3D epithelial cysts. The Caco-2 monolayer is detached to form a single-cell suspension. Cell culture plates are coated in a layer of Geltrex basement membrane extract (BME), onto which the Caco-2 single-cell suspension is added. After 5 days in culture, the single cells grow to form organized multicellular 3D structures, known as cysts or spheroids.

**Step 3**: Immunostaining. Primary antibodies target survivin (a marker of midbodies, a structure present at the end of cell division) and E-cadherin (a cell membrane marker). Secondary antibodies result in the fluorescence of these markers in either green or far red channels.

**Step 4**: Image acquisition. Tile scans are performed with an LSM700 microscope at 4 resolution levels. All 3 channels are captured with each tile scan: Survivin (`CELL0`), E-cadherin or GFP-tubulin (`CELL1`), mCherry-Histone H2B (`CELL2`).

fluorescent protein mCherry. **GFP-tubulin or E-cadherin** (`CELL1`, medium): Tubulin is a marker for microtubules, which are one of the structural components of cells and can be used to see the cell outline/shape. Microtubules also form the mitotic spindle, which is important during mitosis (cell division) for separating DNA into two new daughter cells. Tubulin is tagged with green fluorescent protein (GFP). E-cadherin is a standard marker for epithelial cells. It is found along the cell-cell contact sites and shows epithelial polarity. It is also useful because it stains the membrane of each cell, so this (or GFP-tubulin), in combination with mCherry-Histone H2B shows the cell shape and the nucleus respectively. **Survivin** (`CELL0`, dim): Survivin marks the midbody, which is a bridge between cells present during the very last step of cell division.

`SR-CACO-2` is comprised of more than 9k real pairs of LR and HR patches, per scale, and cell type. It is a representative dataset in this field since it is captured using a standard process in fluorescence microscopy. In addition, three different upscaling scenarios are provided, `X2`, `X4`, and `X8`, in addition to HR, allowing a better study of the limit of SISR methods. The process of capturing our `SR-CACO-2` dataset is described in Sec.2.1, while the experimental protocol described in Sec.2.3 allows preparing the dataset for the design and evaluation of SISR models.

## 2.1 Dataset Capture:

Caco-2 cells (ATCC HTB-37), a colorectal adenocarcinoma cell line, were used for all data collection (Fig.2). These cells were used unmodified, or modified via lentiviral infection for stable overexpression of mCherry-Histone H2B or mCherry-Histone H2B and GFP-tubulin, to label chromosomes and microtubules respectively. Cells were seeded at a density of 12,000 cells per well into μ-slide 8 well plates (Ibidi 80826), pre-coated with 12 μL Geltrex basement membrane extract (Gibco A1413202). Cells were cultured at 37°C in 5% $CO_2$ in Dulbecco's Modified Eagle Medium (DMEM) (Wisent 319-005-CL) supplemented with 10% Fetal Bovine Serum (FBS) (Wisent 091-150, lot 091150) and 2% Geltrex. Media was changed every 2-3 days. Cells were cultured for a total of 5 days to allow the formation of single-layered 3D epithelial structures (cysts) with the open lumen.

After 5 days, cells were fixed in 2% paraformaldehyde in phosphate-buffered saline (PBS) for 10 minutes, followed by immunostaining. Cells were blocked and permeabilized using 10% goat serum, 0.5% fish skin gelatin and 0.5% Triton X-100 in PBS for 1 hour. Primary antibodies were incubated

overnight at 4°C in the blocking/permeabilization buffer. Primary antibodies used were survivin (midbody marker, Cell Signaling 2808T, 1:300 dilution) and E-cadherin (basolateral cell membrane marker, BD Transduction Laboratories 610181, 1:500 dilution). Secondary antibodies were incubated in the blocking/permeabilization buffer for 1 hour at room temperature. Secondary antibodies used were Alexa Fluor 488 AffiniPure Donkey anti-Rabbit IgG (Jackson ImmunoResearch 711-545-152, dilution 1:750) and Alexa Fluor 647 AffiniPure Donkey Anti-Mouse IgG (Jackson Immunoresearch 715-605-151, dilution 1:200). Following immunostaining, cells were stored in PBS at 4°C.

Imaging was performed with a Zeiss LSM700 confocal microscope using a 20x/0.8NA objective lens. Each tile was scanned using a $10 \times 10$ grid at 4 different resolutions. The size of each sub-image in the grid changes with the scale: **HR: 1024x1024 pixels**, scan speed 9, averaging 8; **LR (/2): 512x512 pixels**, scan speed 9, averaging 1; **LR (/4): 256x256 pixels**, scan speed 9, averaging 1; **LR (/8): 128x128 pixels**, scan speed 9, averaging 1. The averaging indicates how many slices are captured and averaged, which reduces noise. Initial images were acquired with HR tile scans completed for each well of cells, before capturing images for subsequent resolutions. Positions were saved using automated Zeiss software, to ensure the same region of interest was imaged at each resolution. Later tiles were acquired with all four resolutions imaged for one well, before imaging subsequent wells.

## 2.2 Dataset Variability:

During the acquisition of `SR-CACO-2`, we ensured the technical and biological diversity of samples. To automate the image acquisition, we programmed the microscope to capture multiple sets of 100 images. Each set of 100 images was stitched together ($10 \times 10$) to form an image tile. Therefore, each tile represents 100 unique images (fields of view) and the 22 tiles correspond to 2,200 unique images. Each image is captured at four scales and for three markers. The automatic collection of a tile of the 4 different resolutions takes over ∼12-16 hours. We note that each tile was captured in a multi-well plate. The 22 wells were collected from 4 independent experiments using 5-6 wells per plate/experiment, ensuring adequate and standard biological diversity.

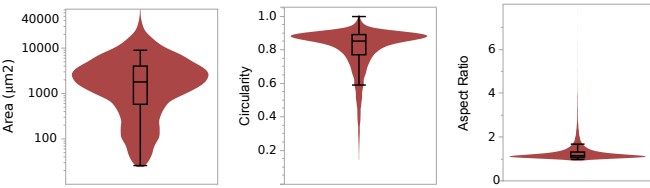

Figure 3: Object-based analysis of cellular structures captured of all the 2,200 high-resolution images ($22 \times 10 \times 10$).

To further assess the diversity of our dataset, we performed an object-based analysis of cellular structures captured in the image dataset. The 2,200 ($22 \times 10 \times 10$) unique image fields of view at the high resolution contain ∼ 16,800 multi-cellular objects. The cells were cultivated in a three-dimensional protein matrix that allows them to form tissue-like cyst structures that resemble the natural organization of cells in tissues and organs, such as the colon, lung, breast, etc. Cells grown in a three-dimensional matrix exhibit spontaneous variations in cyst size and organization (hollow or solid). We recently [87, 86] reported multiple cellular phenotypes in Caco2 cells that resemble tissue geometries found in healthy and disease states. Our dataset contains cysts with size variations over 2 orders of magnitude. Moreover, shape analysis of the cysts demonstrates substantial variation in aspect ratio and circularity. This reflects cysts covering a spectrum of phenotypes, including single-layered, multi-layered, and solid. This contrasts with cells grown as two-dimensional cultures on plastic or glass surfaces, where there is substantially less variation in size and organization when viewing single cells or confluent patches as is often used in other recently published microscopy datasets. Fig.3 shows the analysis of the cells in the `SR-CACO-2` dataset. We can observe from these plots that the cells span a large size range with a relatively circular shape.

Our dataset also includes diverse cellular markers with unique properties. Cells were labeled with Hoechst dye that labels chromatin and is frequently used as a nuclear stain in cell biology. We also include mCherry-Histone, a red-fluorescent protein conjugated to histone H2B that is part of chromatin. This serves as an additional marker of cell nuclei and represents a signal that is often employed in live imaging experiments. mCherry-Histone represents a lower signal than Hoechst

because the cells were selected to have a low-to-moderate expression of this protein and are detected by direct fluorescence. The other two markers (E-cadherin and Survivin) represent typical staining obtained with indirect fluorescence methods, using primary and fluorophore-conjugated secondary antibodies. E-cadherin is expressed in all Caco2 cells and is a cell surface protein. Therefore, this has a distinct expression pattern from the blob-like nuclear labels described above. Despite being expressed in all cells, there is heterogeneity at the cellular level, providing extensive natural variation within our dataset. Survivin is transiently expressed in cells during cell division. This represents an additional unique marker pattern that appears as an intense patch connecting two cells. Survivin is only expressed in a subset of cells during mitosis and cytokinesis, processes that often utilize imaging and machine learning for analysis [40].

## 2.3  Experimental Protocol:

**Patches sampling.** Since the imaged tiles are quite large ( $9k \times 9k$ ), they are not well suited for machine learning models as they usually operate on smaller images. To this end, we pre-processed the tiles to patches as presented in Fig.4. On HR tiles of the brightest cell, *i.e.* CELL2, a sliding window was employed that scans from left to right and from top to bottom of the entire tile. Windows can overlap by $25\%$ to increase the chance of capturing large patterns and minimize cropping cells. The window size is set to $576 \times 576$ with $64$ extra margin at each side that will be useful later for registration. This additional margin was discarded, and only windows of size $512 \times 512$ were preserved as final patches.

Many of the sampled windows will land on empty regions, *i.e.*, black background. Across all the 22 HR tiles of CELL2, $74.24\%$ of total *pixels* are background while only $25.74\%$ are foreground (cells). Windows which are more black will not be helpful in learning or evaluation since they are easy to reconstruct using SISR models. They would introduce an evaluation bias by giving a high level of performance, which does not reflect the true performance of the models to super-resolve relevant regions, *i.e.*, cells. For this reason, we discarded the windows without enough cell content. To detect these windows, we apply a threshold to tile with a value of $4$ which is deemed sufficient to spot the cells without including noise. Only windows with at least $20\%$ of cell content were preserved.

The aforementioned process is performed over HR tiles of CELL2 to determine useful HR patches. Once done, the coordinates of these patches are used to crop HR patches from HR tiles of CELL1 and CELL0 at the same location. Then, the coordinates of LR patches from LR tiles are estimated using the coordinates of the corresponding HR patches. This is performed by downsizing the coordinates using the corresponding downscale factor. For example, for the LR patch at scale /2 for CELL2 in a specific location, we divide the coordinates of its corresponding HR patch of the same cell and location by 2.

Scanning the same tiles at different resolutions can lead to small but detectable shifts (fractions of a micron) due to the physical movement of the microscope stage from the position of tile 100 back to the imaging start position at tile 1. This is a common issue for any type of microscope with a mechanical stage. To mitigate this issue, we align patches once cropped. Given an HR patch and its corresponding LR patches, we perform image registration of each LR patch to be aligned with its corresponding HR patch, where the HR patch is used as the image reference. A *global* shift between the two patches is estimated using TV-L1 algorithm for optical flow estimation [101][1]. This allows performing a global shift of all the pixels at once while preserving all their neighbouring pixels. Such a shift can lead to artifacts at the patch boundaries. Therefore, we re-crop the patch at the center with size $512 \times 512$ and discard the extra margin of $64$ at each side. The final HR patches are $512 \times 512$ while LR patches of /2, /4, and /8 are $256 \times 256$, $128 \times 128$, and $64 \times 64$, respectively.

**Data split.** First, the 22 tiles are randomly split into 15 for training, 3 for validation, and 4 for testing. This prevents mixing patches of the same tile across different sets. Tiles with identifiers (ID) = ['9', '10', '14', '20'] are assigned to the test set. Tiles with ID ['7', '11', '19'] are used for the validation set. Finally, tiles with ID = ['1', '2', '3', '4', '5', '6', '8', '12', '13', '15', '16', '17', '18', '21', '22'] are used for training. This is summarized in Tab.1 along with the patch distribution for all subsets.

---

[1]In practice, we used the function `skimage.registration.optical_flow_tvl1` from scikit-image library `https://scikit-image.org`.

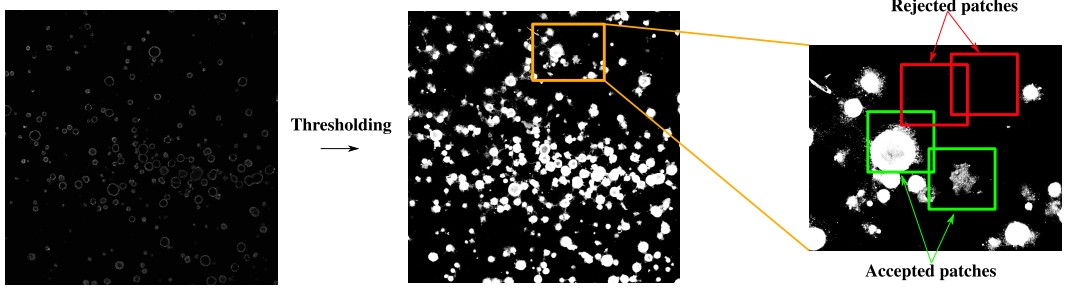

Tile, HR, CELL2

Figure 4: Pre-processing of tiles to patches. First, the HR tile of cell CELL2 (the brightest) is binarized to allow localizing cells only (*i.e.* ROI). A sliding window of size $512 \times 512$ is used to scan the entire tile with specific overlap. Only windows with enough cell mass are preserved while the rest are discarded. The same coordinates of the preserved patches are used for CELL1 and CELL2 of HR tiles. Coordinates of LR patches X2, X4, and X8 are computed by scaling down the HR patch coordinates according to the corresponding scale.

| | Tiles | | | | Patches | | | |
|---|---|---|---|---|---|---|---|---|
| Data subsets | Train | Validation | Test | Total | Train | Validation | Test | Total |
| Number | 15 | 3 | 4 | 22 | 7,349 | 1,117 | 1,471 | 9,937 |
| Image size: | | | | | | | | |
| HR | | $\sim 9,318 \times 9,318$ | | | | $512 \times 512$ | | |
| LR /2 | | $\sim 4,658 \times 4,658$ | | | | $256 \times 256$ | | |
| LR /4 | | $\sim 2,328 \times 2,328$ | | | | $128 \times 128$ | | |
| LR /8 | | $\sim 1,164 \times 1,164$ | | | | $64 \times 64$ | | |
| File size: | | | | | | | | |
| HR | | 260.5 MB | | | | 262.4 KB | | |
| LR /2 | | 65.1 MB | | | | 65.8 KB | | |
| LR /4 | | 16.3 MB | | | | 16.6 KB | | |
| LR /8 | | 4.1 MB | | | | 4.4 KB | | |

Table 1: Split of SR-CACO-2 into the train, validation and test subsets, along with relevant statistics. Numbers are defined per scale (X2, X4, X8, and HR), and per cell type (CELL0, CELL1, and CELL2).

## 3 Single-Image Super-Resolution Methods

In addition to the proposed dataset, we provide an extensive benchmarking of different SISR methods. In particular, 16 representative SISR methods were selected from 4 common families [92] are evaluated. We also compare to a simple baseline which is interpolation via the bicubic method. Our results can serve as baselines for future comparisons. We selected the following state-of-the-art methods as follows. **Pre-upsampling SR**: SRCNN [19], VDSR [42], DRRN [77], and MemNet [78]. **Post-upsampling SR**: NLSN [59], DFCAN [65], SwinIR [55], EDSR (LIIF) [16], ENLCN [95], GRL [52], ACT [99], and Omni-SR [85]. **Iterative up-and-down sampling SR**: DBPN [28], and SRFBN [54]. **Progressive upsampling SR**: ProSR [89], and MS-LapSRN [44]. The post-upsampling family is the most recent and dominant approach [92]. Other families have seen less progress mainly due to their high computational cost.

Given the limited space, we provide the training details, ablations, evaluation metrics, and more results in the supplementary materials. In summary, we follow a standard performance evaluation protocol for SISR. Three standard measures were used to evaluate the performance of SISR methods – peak signal-to-noise ratio (PSNR), structural similarity index (SSIM) [91], and normalized root mean square error (NRMSE) [65]. The evaluation is performed on the full patch, referred to as *full image*, as commonly done in the literature. Moreover, we report refined performance over the cells only, referred to as *ROI only*, to assess the predictive quality without the black background inside the patches. ROIs are determined by thresholding the HR patch using a set of thresholds[2]. Performance is computed per threshold over the ROI only. The final performance is reported as the average of all per-threshold performances. SISR models are trained on each cell type, and each scale separately.

**Super-Resolution Task**: Table 2 shows results on ROI only. Given the space limitations, we present only one method per family. The full results are in the supplementary materials. Overall, SISR

---

[2]Evaluation thresholds are $\{4, 5, 6, 7, 8, 9, 10\}$.

methods achieved better results than simple interpolation. As commonly known, the higher the scale factor, the more difficult the task. This is also observed in our results across all methods and performance measures. For instance, SRCNN [19] yields a PSNR of 35.08, 33.17, and 29.19 for X2, X4, and X8, respectively. Another emerging pattern in these results is the discrepancy performance across cell types: CELL0, CELL1, and CELL2. Over PSNR and SSIM performance, CELL0, *i.e.* dimmest, is the easiest while CELL2 is the most difficult, *i.e.* the brightest. Over NRMSE, results are mixed. Across all four studied SISR families, we observe that pre-upsampling SR yielded better results overall across all metrics and scales. This is unexpected since post-upsampling SR are state-of-the-art approaches [92]. These results may suggest that *upscaling* our LR images through deep models may fail to reconstruct the details of our HR images. This may be explained by the nature of the details present in our HR images which paradoxically comes from noise . Confocal microscopes perform multiple scans to produce a set of slices that are aggregated to produce a final 2D image. Due to an inherited noise produced by the detectors on microscopes, each slice is different. However, averaging these slices decreases the noise and maintains the true signal. The work of [83] shows that it is difficult for CNN-based models to reconstruct input images with noise as they tend to filter out that noise even when it is provided as input. Therefore, it is challenging for such models to produce similar images to our HR images. Such results should be considered in designing future SISR methods for this dataset. We provide in Fig.5 typical visual results for scale X2 over CELL2 which illustrate the limited capacity of these methods to produce accurate HR details as they yield blurry images.

In some cases, the methods fail, yielding a decline in performance such as the case of MS-LapSRN [44]. We have analyzed the reason for this decline in performance and found that the model fails to converge properly as the loss does not decrease as expected. This is most clear on CELL0 data leading to a large decline compared to CELL1 and CELL2. We believe that this is caused by the combination of the very low level of brightness for the cell, and the use of residual learning in MS-LapSRN. This model has difficulty producing very small residuals. Instead, it yields large residuals. This can be confirmed when measuring performance over a full image, i.e., by including a dark background where the typical intensity is 0, which leads to a sharp decline in performance. By inspecting some predicted samples, the intensity of these background regions is typically 10. Note that this is not the case for the ProSR model that also uses residual learning. However, ProSR [89] employs a different architecture than MS-LapSRN which, in this case, is much more efficient due to its depth or its local residual layers. During our experiments, we observed that hyper-parameters do not transfer well across cell types or scales of a model. Therefore, we performed a grid search separately for each model, for each cell type, and each scale. This provides each method with a good and fair chance to perform well. Failure cases are mostly related to the method and the data's nature.

| SISR Methods | Scale | PSNR ↑ | | | | NRMSE ↓ | | | | SSIM ↑ | | | |
|---|---|---|---|---|---|---|---|---|---|---|---|---|---|
| | | CELL0 | CELL1 | CELL2 | Mean | CELL0 | CELL1 | CELL2 | Mean | CELL0 | CELL1 | CELL2 | Mean |
| Bicubic | X2 | 35.02 | 32.15 | 30.38 | 32.52 | 0.1085 | 0.0601 | 0.0724 | 0.0803 | 0.7618 | 0.7658 | 0.6891 | 0.7389 |
| | X4 | 35.46 | 32.03 | 31.10 | 32.86 | 0.0985 | 0.0586 | 0.0660 | 0.0744 | 0.8206 | 0.8002 | 0.7673 | 0.7960 |
| | X8 | 31.88 | 27.50 | 26.10 | 28.49 | 0.1655 | 0.1139 | 0.1349 | 0.1381 | 0.6683 | 0.6266 | 0.6511 | 0.6487 |
| **Pre-upsampling SR** | | | | | | | | | | | | | |
| SRCNN [19] *(eccv,2014)* | X2 | 37.54 | 34.27 | 33.42 | 35.08 | 0.0710 | 0.0450 | 0.0500 | 0.0553 | 0.8517 | 0.8524 | 0.8210 | 0.8417 |
| | X4 | 36.14 | 32.73 | 32.25 | 33.71 | 0.0817 | 0.0528 | 0.0572 | 0.0639 | 0.8522 | 0.8216 | 0.8079 | 0.8272 |
| | X8 | 33.05 | 28.04 | 26.49 | **29.19** | 0.1265 | 0.0967 | 0.1220 | **0.1151** | 0.7711 | 0.7085 | 0.7092 | **0.7296** |
| **Post-upsampling SR** | | | | | | | | | | | | | |
| Omni-SR [85] *(cvpr,2023)* | X2 | 37.70 | 34.11 | 33.51 | 35.11 | 0.0759 | 0.0461 | 0.0496 | 0.0572 | 0.8744 | 0.8539 | 0.8313 | 0.8532 |
| | X4 | 36.44 | 32.59 | 32.34 | 33.79 | 0.0849 | 0.0536 | 0.0563 | 0.0649 | 0.8592 | 0.8203 | 0.8111 | 0.8302 |
| | X8 | 30.75 | 27.16 | 25.30 | 27.74 | 0.1713 | 0.1098 | 0.1352 | 0.1387 | 0.6715 | 0.6419 | 0.6591 | 0.6575 |
| **Iterative up-and-down sampling SR** | | | | | | | | | | | | | |
| SRFBN [54] *(cvpr,2019)* | X2 | 36.13 | 33.15 | 31.61 | 33.63 | 0.0955 | 0.0531 | 0.0625 | 0.0704 | 0.8078 | 0.8091 | 0.7470 | 0.7880 |
| | X4 | 36.08 | 32.52 | 31.79 | 33.46 | 0.0911 | 0.0545 | 0.0605 | 0.0687 | 0.8405 | 0.8147 | 0.7889 | 0.8147 |
| | X8 | 32.27 | 27.78 | 26.47 | 28.84 | 0.1560 | 0.1091 | 0.1278 | 0.1310 | 0.7022 | 0.6549 | 0.6904 | 0.6825 |
| **Progressive upsampling SR** | | | | | | | | | | | | | |
| MS-LapSRN [44] *(tpami,2019)* | X2 | 33.88 | 32.36 | 29.34 | 31.86 | 0.1130 | 0.0535 | 0.0791 | 0.0819 | 0.7652 | 0.8164 | 0.7695 | 0.7837 |
| | X4 | 30.80 | 30.99 | 31.08 | 30.96 | 0.1192 | 0.0615 | 0.0626 | 0.0811 | 0.7885 | 0.7837 | 0.7806 | 0.7843 |
| | X8 | 31.83 | 27.14 | 25.06 | 28.01 | 0.1404 | 0.0982 | 0.1323 | 0.1236 | 0.7478 | 0.6933 | 0.6640 | 0.7017 |

Table 2: The performance of SISR methods on the SR-CACO-2 test set of ROI only, *i.e.*, cells. **See supplementary materials for the full 16 methods.**

**Downstream Biology Tasks – Object Detection/Segmentation.** To evaluate performance from each SISR model on downstream biology tasks, we evaluated each model at X2, X4, and X8 using an object segmentation problem that is widely used and openly accessible software for image analysis

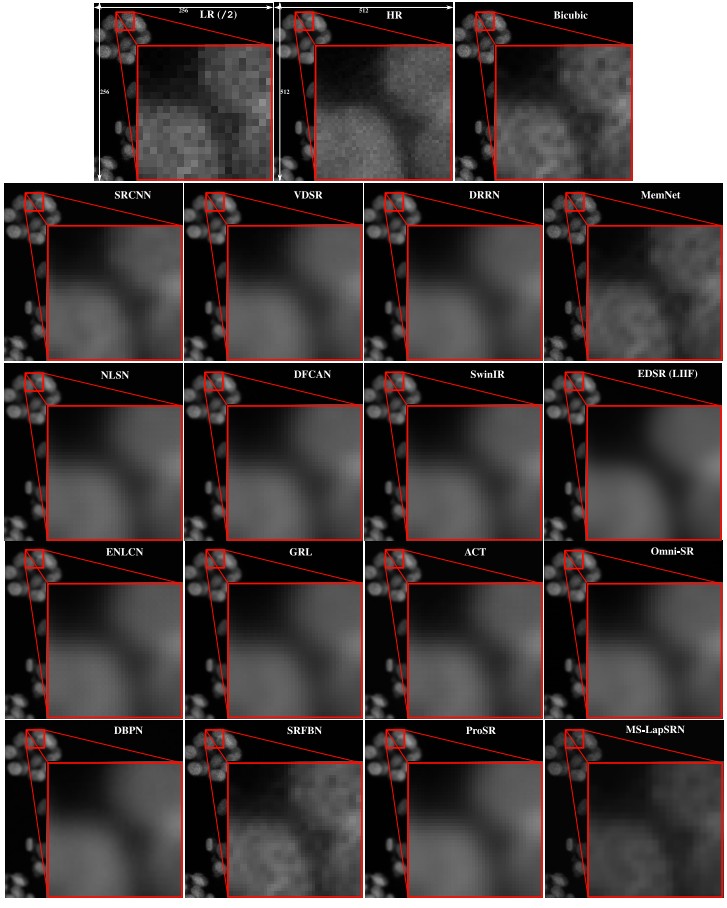

Figure 5: Illustrative visual results for X2, CELL2 across all SISR models. HR patch file sample name: `hr_div_1/tile_HighRes1024-10_293_6912_7552_6400_7040_CELL2.tif`.

(ImageJ) [71][3]. 30 random images from the test set are selected. Images were resized to 512 x 512 pixels without interpolation. Automatic object segmentation was performed using Star-convex Shapes (StarDist) [70, 93, 94] with the following parameters: modelChoice = Versatile (fluorescent nuclei), normalizeInput = TRUE, percentileBottom = 1.1, percentileTop = 100.0, probThresh = 0.6, nmsThresh = 0.4. The resulting output was the number of objects detected. Manual object counting was performed to create the Ground Truth number of objects. Manual comparisons were made to identify over-segmentation and under-segmentation errors for each image. Graphing and statistical analysis were performed using JMP Pro (version 17.0.0)[4]. Steel's test for non-parametric multiple comparisons using HR as the control group was performed to assess model performance. Our objective was to evaluate model performance relative to HR, not ground truth. Models that perform as well as HR (i.e. not statistically different than HR at p=0.05) are represented in orange. Fig.6 shows the results for CELL2, X2 where the VDSR, DRRN, NLSN, DFCAN, ENLCN, ACT, Omni-SR, DBPN, and ProSR models show promising results. The supplementary materials provide more analysis.

## 4 Ethical Considerations and Dataset Accessibility

The SR-CACO-2 dataset does not require ethics approval. Caco-2 (*Ca*ncer *co*li-2) cells were isolated from a 72-year-old white male in the 1970s as part of a collection of gastrointestinal cancer cell lines at the Memorial Sloan Kettering Cancer Center [46]. The use of Caco-2 cells does not require ethics approval because the cells were established prior to the US Federal Policy for the

---

[3]https://imagej.net/ij

[4]https://www.jmp.com/en_ca/home.html

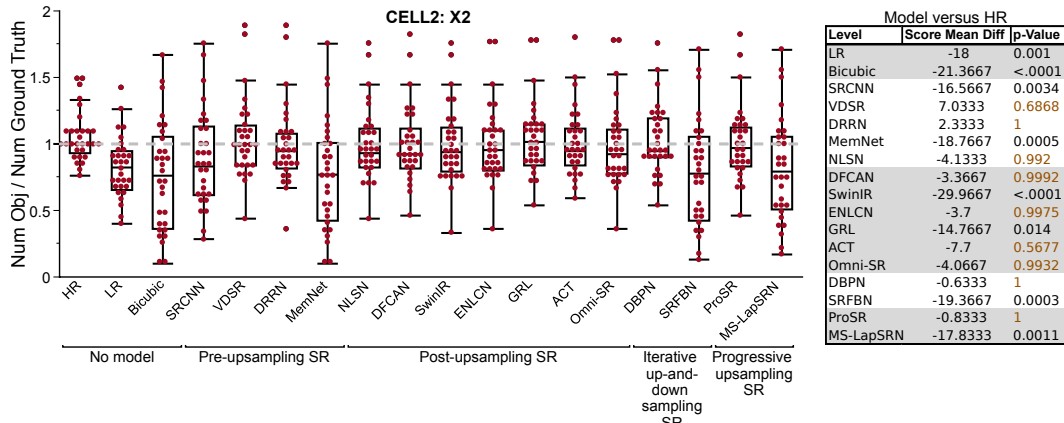

Figure 6: Analysis of cell detection performance for `CELL2`, `X2` over 30 random test samples (red dots). More results can be found in the supplementary materials.

Protection of Human Subjects (1991) and the UK Human Tissue Act (2004). This is the standard for the biological field for the use of cell lines derived from human tissue that are now considered publicly available. The cells were obtained from the American Type Culture Collection (ATCC)[5], a nonprofit organization that collects and distributes cell lines and other biological materials that are publicly accessible. Its legal disclaimers allow for the use of Caco-2 cells for laboratory research https://www.atcc.org/products/htb-37.

The dataset is made public and can be freely accessed under the Creative Commons Attribution-NonCommercial-ShareAlike 4.0 International license (CC BY-NC-SA 4.0) https://creativecommons.org/licenses/by-nc-sa/4.0/. The dataset which includes full tiles, and patches, is stored permanently on Google Drive. The provided code uses an open-source license and it is located at https://github.com/sbelharbi/sr-caco-2 along with the dataset downloading link. The pretrained weights can be found at https://huggingface.co/sbelharbi/sr-caco-2.

## 5  Conclusion

The few existing datasets for SISR microscopy are mostly private which limits research advancements. Our work seeks to address this gap. In particular, we propose a new dataset, `SR-CACO-2`, for Confocal Fluorescence Microscopy Image Super-Resolution. It contains 3 scales (`X2`, `X4`, and `X8`, in addition to HR) with three protein markers in different lighting conditions and about 9k real pairs of LR/HR patches. The procedure for data capture and the experimental protocol for the evaluation of SISR methods are follow standard practices, in addition of being reproducible. Our benchmarking of state-of-the-art SISR methods indicates that they cannot accurately produce the HR images, making this new dataset extremely challenging. Aside from the evaluation of quality for super-resolved images, we conducted further analysis to assess their performance in downstream biology tasks such as cell segmentation. Several methods achieved promising results compared to LR and HR images over cell detection/segmentation tasks. `SR-CACO-2` can contribute to driving progress in the design of SISR models for microscopy. This extends from fixed-imaging to live-imaging. SISR models trained with our dataset can be employed for live-imaging videos captured with low quality under reduced light exposure to preserve cells. This leads to fast imaging which allows for the observation of instantaneous inter-cellular events with less damage to the cells. Accurately upscaling these videos with SISR models may ultimately open the door to performing biological tasks such as cell tracking, counting, and segmentation.

**The following supplementary materials** contain: the related works, a discussion on the limitations of `SR-CACO-2` dataset, an empirical comparison between interpolated and microscopic LR images, an overview of `SR-CACO-2` dataset (more statistics and data format), a hosting/licensing/intended uses/ethical considerations, SISR baselines with the evaluation protocol, implementation details, and more results.

---

[5]https://www.atcc.org

## Acknowledgments and Disclosure of Funding

This research was supported in part by the Canadian Institutes of Health Research, the Natural Sciences and Engineering Research Council of Canada, and the Digital Research Alliance of Canada.

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

**Part**

# Supplementary Materials

## Table of Contents

## A  Related Work

**1. Photo-realistic SISR**: A large progress has been observed in SISR for photo-realistic application [92, 96]. Earlier to deep learning methods, SISR methods pursued different approaches [68]. For instance, image statistical methods leverage different priors about the image such as minimal local variation by exploiting total variation [23], maximum entropy [64], and other priors [22]. Another dominant and successful technique is example-based approach [92, 96] where an image is decomposed into patches, and the super-resolved image is obtained by upscaling each patch. These methods typically rely on dictionaries and sparse coding [82, 102].

The introduction of deep SISR [19, 20], changes they way to tackle this task. The proposed methods mostly deal with supervised SISR task. Commonly, the degradation process is known beforehand and often interpolation is used to synthesize LR images from HR images. In the following, we divide deep super-resolution (SR) existing works into four main families based on their framework [92] which differs in the way the SR image is upscaled based on the input LR image which is the main key problem in SISR task (Fig.7). In particular, each family differs in what type of upscaling is used and its location in the model.

*Pre-upsampling super-resolution.* Initial works circumvented the difficulty of mapping LR to HR images by performing an initial upscaling step using standard algorithms such as interpolation which is then refined using deep learning models. SRCNN [19, 20], one of the pioneering works in SISR, firstly adopted such approach where bicubic interpolation is initially used to upscale the input image, followed by a convolutional neural network (CNN). This CNN is trained end-to-end where it takes the upscaled input and aims to refine it, in a supervised way, to yield a SR image as close as possible to the HR image. A major advantage of these methods is that they only need to focus on the refinement since a major part of the SISR has already been done by interpolation. Other works followed similar approach with differences in training loss or architecture design [41, 42, 77, 78]. For instance, VDSR [42] employs residual learning as loss where the model aims to learn the residuals between the bicubic and the HR images, in addition to using a deeper model than SRCNN. Other methods often rely on deep models, in particular using recursion such as DRCN [41], DRNN [77], and MemNet [78]. Despite the success of these methods, they are very computationally expensive and they require a lot

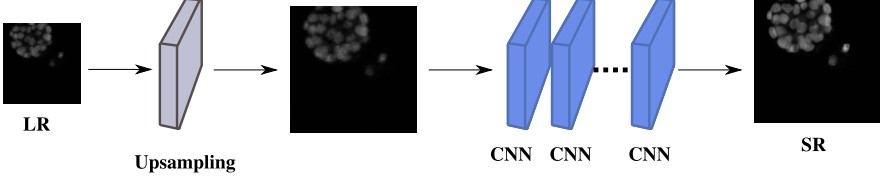

**Pre-upsampling SR**

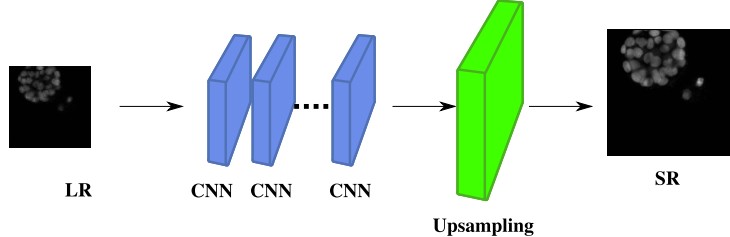

**Post-upsampling SR**

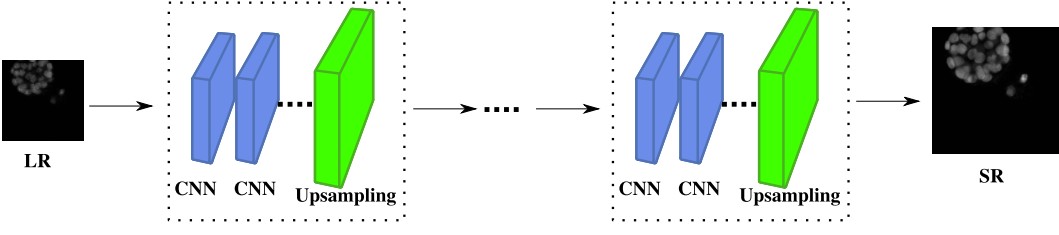

**Progressive upsampling SR**

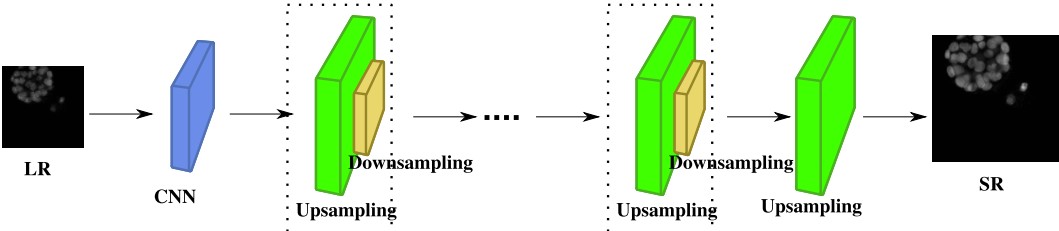

**Iterative up-and-down sampling SR**

Figure 7: SISR families. The grey block indicates a predefined upsampling such as interpolation. Blue, green, and yellow indicate learnable convolution, upsampling, and downsampling modules, respectively.

of memory since these very deep models perform all their operations in HR image space. Subsequent works aim to find different approaches to bypass this computational cost while improving the quality of the produced SR image.

*Iterative up-and-down sampling super-resolution.* This family leverages back-projection technique [37] suggested in [80] as a way to improve SR methods. In particular, it is used in an iterative way to reinforce the mutual dependency LR-HR in image pairs. For instance, DBPN [28] uses consecutive up and down layers. A residual error is then added to current image prediction which is then feed to the next layer. The final layer concatenates all the previous predictions, then, followed by a CNN layer which yields the final prediction. A parallel work, SRFBN [54], uses a similar strategy with more dense skip connections. Such approach has been also used in video super-resolution [29] but it has not been adopted largely by the community.

*Progressive upsampling super-resolution.* This family of methods tackles two common issues in SISR. Exiting works upscale the LR image at once which is difficult. A second issues is that each scale requires training one model. This category aims to build a single model for all scales, while scaling up the LR image gradually. This creates less cumbersome models for deployment, and also allows multi-step upscaling which facilitates more SISR task. For instance, LapSRN [45] and MS-LapSRN [44] adopted Laplacian pyramid SR network. It is based on a cascade of CNN that learns residual sub-bands of high resolution images, which is added to an interpolated version of the image. This is done repeatedly where the scale increases until reaching the final requested scale where the interpolated images has accumulated all the residuals. ProSR [89] follows a similar approach however it uses the same input for interpolation instead of intermediate resulting image. The methods under this family greatly reduce the learning difficulty by decomposing the task into multiple steps that support each others. However, training models with multi-stage scales remains difficult and vulnerable to instability.

*Post-upsampling super-resolution.* These methods perform most of the computations in LR dimensions while placing learnable upsampling layers toward the end for minimal computations at HR dimensions. The aim is to reduce the training computational cost which is a well known bottleneck in SISR [21, 73]. SISR field is dominated by this framework. Several works have been proposed. FSRCNN [21] improved SRCNN [19] by changing its architecture from a pre-sampling method to post-sampling one. This is done by introducing a deconvolution layer at the output layer while learning an end-to-end mapping from LR to HR space. To furthermore reduce computation, they shrink feature maps towards the input then expand them back toward the last layers while using small filter size but more mapping layers. Different methods focused on improving residual learning for deep models as it has been shown to be beneficial for SISR task, including EDSR [56], RCAN [103], RDN [105], CARN [2], and MSRN [50]. In SRDenseNet [81], authors leverage more skip connections in a very deep network. Adversarial generative models have also been introduced in SRGAN [47] and ESRGAN [88]. Non-local aggregation method has been extensively used in image restoration. Under the assumption that similar patches frequently recur in a natural scene image, several SISR methods have been proposed to leverage this idea. IGNN [106] extends the Non-local self-similarity idea across scales using graphs. Attention-based models have been also developed for SISR task to improve features such as RNAN [104], SAN [17], HAN [62], SwinIR [55], NLSN [59], ENLCN [95]. In GRL method [52], authors propose to explicitly model image hierarchies in the global, regional, and local range into a new self-attention module. In ACT method [99], authors improve self-attention by including local features from CNNs and long-range multi-scale dependencies captured by transformers. Omni-SR method leverages both spatial and channel-wise self-attention [85]. Recently, iterative-based approaches such as diffusion methods have attracted the community attention [67, 100]. However, they remain computationally expensive.

A recent line of research tackles the task of arbitrary-scale SR (ASSR) [16, 24, 30, 39, 53, 57, 98] where the aim is to design techniques that can upscale an image to any arbitrary scale, mainly by relying on neural fields [12, 14, 48] to to represent continuous signals that can be sampled at arbitrary rates. For instance, in [16], authors employ local implicit neural representation to represent images in a continuous domain. In particular, they propose a Local Implicit Image Function (LIIF) method. ASSR is achieved by replacing standard fixed-upsampling techniques with a multilayer perceptron (MLP) used to query pixel-value at any coordinate and any scale.

**2. Microscopy SISR**: Compared to photo-realistic field, microscopy domain has seen very limited work. Each existing work is specialized in one single imaging technique. Some of these methods enhance image quality [8, 35, 60, 63], while others aim to upscale images by a factor [51, 65]. In [8], authors propose task-assisted generative adversarial network (TA-GAN) which incorporates an auxiliary task such as segmentation which has shown to be better than unassisted methods. It has been used to convert confocal to STED images. Authors in [35] perform a similar task using GANs. Deep models haven used in [60] for STORM modality while PALM modality has been tackled in [63]. Additionally, recurrent neural networks (RNN) have been used to upscale images for STORM modality [51]. Authors in [65] propose deep Fourier channel attention network for SIM modality. In [58], authors consider a GAN-based approach SISR of cytopathological images. All these works employ private real or synthetic datasets. Only the work of [65] provides a public dataset for SIM modality. Such lack of public data can contribute in slowing down research in microscopy SISR field. Our work is an effort to provide a public dataset for fluorescence microscopy SISR task for model's training and evaluation.

It is worth mentioning that there is an emerging line of research in SISR for natural scene images that deal with real-world SISR [11, 13, 38, 49, 97]. Such methods avoid using simulated low resolution images obtained by interpolation. Instead, real low resolution images are used in order to deal with real-world scenarios.

## B `SR-CACO-2` Dataset Limitations

First, it would have been interesting to supplement this data set with oversaturation/undersaturation or very bright vs very dim imaging, as live imaging of cells requires low laser power to minimize damage to cells. This therefore often results in dim images, compared to what can be acquired using fixed and immunostained images. Second, Caco2 cells were ideal for cell for use in this study due to the ease and efficiency of culturing them in 3D and their ability to efficiently polarize. However, often the next step in biological studies is to use human patient-derived tissue or mouse tissue. Future studies could involve the imaging of either primary human or primary mouse organoids. Third, live cells are more dynamic, while fixed cells are static. Live cells are actively growing, dividing, and synthesizing building blocks. This results in changes to the markers that we have used, like mCherry-Histone H2B and GFP-tubulin, which can change shape and position within an individual cell over time, and provide important biological information, such as the stage of cell division. This information cannot be followed for an individual cell once fixed and immunostained, which may impact the SISR model performance. Although, live cells look nearly identical to fixed cells, using fixed dead cells for training SISR models remains a practical solution. Ideally, collecting images of live cells as time-lapse videos for training could be better choice as it reflects the test time scenario. However, acquiring such data is extremely challenging for the task of SISR due to cell movements or appearance of new cells due to cell division. This will create a large misalignment between tiles at different scales, rendering the training impractical. Forth, the dataset is limited in term of the types of proteins that have been captured. We are only looking at chromosomes, cell membrane/cell structural proteins, and proteins specific for division. The dataset could perhaps be improved with an increase in the number and type of markers used, such as proteins that appear as puncta in the cytoplasm, proteins localized to the apical membrane (mutually exclusive localization with E-cadherin), basement membrane proteins (extracellular matrix found around the outside of cysts), and signaling molecules.

## C Low Resolution Images: Interpolation vs Microscope

We conducted an empirical analysis to compare both images: real (microscopic) LR vs. bicubic LR. Here are some relevant observations:

- Pixel-intensity difference: both LR types are different. Their absolute pixel-wise difference value can be in [0, 50] as well as [200, 255] (see the |REAL - BICUBIC| histogram in Fig.8). The difference is mainly concentrated over cell regions.

- Compared to real LR, bicubic LR can maintain better cell structure and even full cells from the HR images (see the real and bicubic images in Fig.8).

- Intensity span: depending on the image, the real LR images are sometimes much more expressive in terms of pixel intensity as they span larger intensities compared to bicubic LR (see the pixel-intensity histogram in Fig.8).

- Real LR images are more noisy compared to bicubic LR ones which tend to be very smooth (see the visual result of the Laplace filter, and the histogram of its absolute value |Laplace LR| in Fig.8). Note that the noise in real LR is the results of a single scan while HR are scanned 9 times to be averaged.

This large contrast between real and bicubic LR only confirms that bicubic LR cannot be used as a replacement to substitute real LR images for training and evaluation, as done in SR methods over natural scene images. Real LR images must be used to effectively simulate a realistic scenario for the model at deployment time. Therefore our collection of real LR images is extremely valuable to designing realistic SR models.

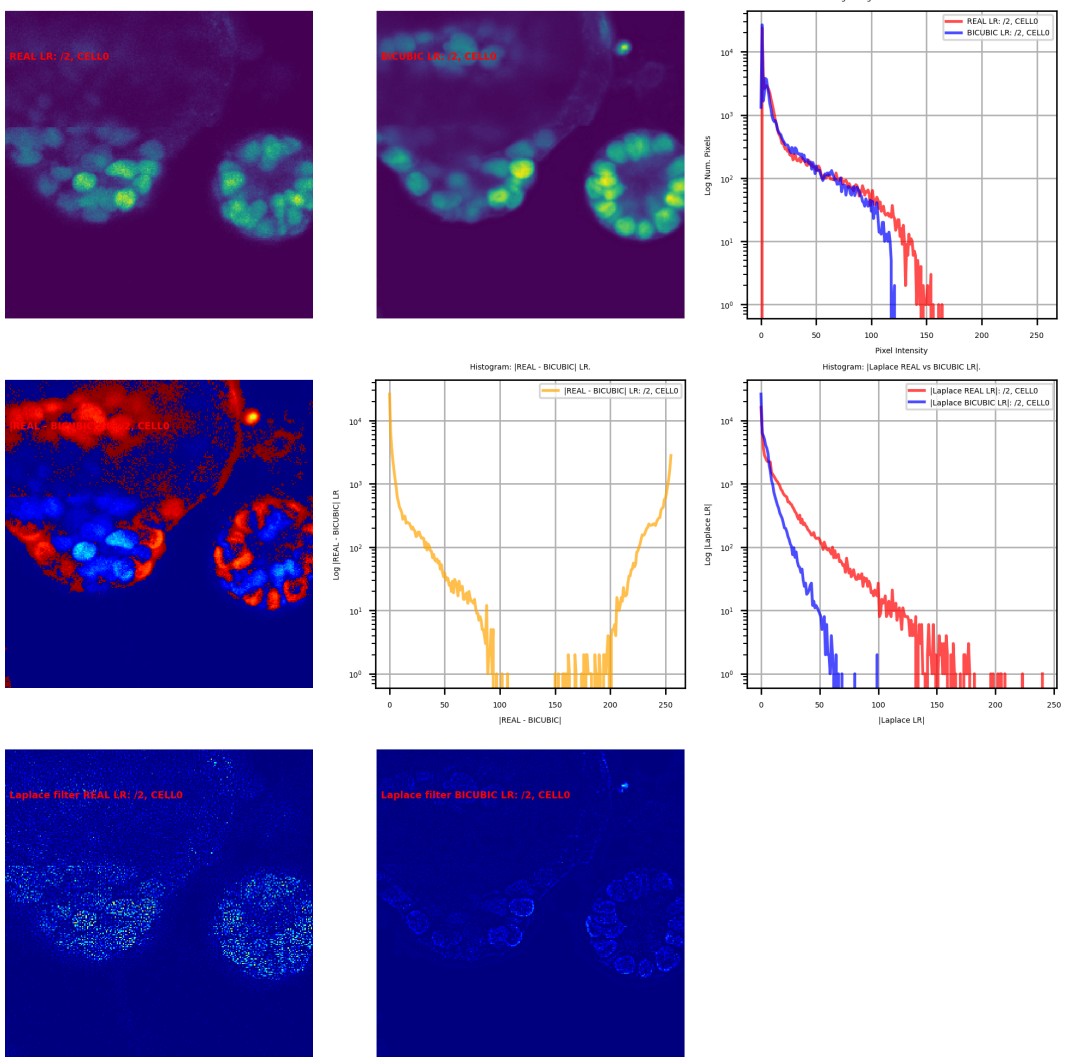

Figure 8: Interpolated vs Microscopic low resolution image for scale /2 and CELL0. For interpolation, we used bicubic interpolation of HR image to obtain low resolution with down scale factor of 2.

# D   Overview of Dataset

## D.1   Dataset statistics

We provide a nutrition label for our `SR-CACO-2` dataset (Fig.9). Furthermore, Tab.3 presents more statistics about ROI distribution per HR tile for `SR-CACO-2` dataset. Note that cells cover only small part of a tile in general.

## D.2   Data Format and Organization

Figure 10 shows the file organization of patches and tiles for `SR-CACO-2` dataset. The names of the tiles follows this format:

1. HR tiles: `HighRes1024/HighRes1024-<ID>.tif`

2. LR tiles (/2): `LowRes512/LowRes512-<ID>.tif`

3. LR tiles (/4): `LowRes256/LowRes256-<ID>.tif`

4. LR tiles (/8): `LowRes128/LowRes128-<ID>.tif`

# SR-CACO-2 **Dataset Facts**

**Dataset** SR-CACO-2
**Nature of Dataset** A Dataset for Confocal Fluorescence Microscopy Image Super-Resolution
**Unique images per Dataset** 2,200
**Tiles Per Dataset** 22 (each tiles is $10 \times 10$ unique image)
**Patches Per Dataset** 9,937 (per scale, per cell-type)
**Available scales** HR, LR (X2, X4, X8)
**Available markers** CELL0 (Survivin), CELL1 (E-cadherin or GFP-tubulin), CELL2 (mCherry-Histone H2B)

---

Motivation

---

**Summary** A Dataset for Confocal Fluorescence Microscopy Image Super-Resolution. It is comprised of low- and high-resolution image pairs marked for three different fluorescent markers. It allows the evaluation of performance of SISR methods on three different upscaling levels (X2, X4, X8). SR-CACO-2 contains the human epithelial cell line Caco-2 (ATCC HTB-37), and it is composed of 22 tiles that have been translated in the form of 9,937 image patches for experiments with SISR methods.
**Original Authors** Soufiane Belharbi, Mara KM Whitford, Phuong Hoang, Shakeeb Murtaza, Luke McCaffrey, Eric Granger

---

Metadata

---

**URL** https://github.com/sbelharbi/sr-caco-2
**Keywords** Confocal Fluorescence Microscopy, Image Super-resolution, Deep Learning, Benchmark
**Cell type** Human epithelial cell line Caco-2 (ATCC HTB-37)
**Protein markers** Survivin, E-cadherin or Tubulin, and Histone H2B
**Format** .tif
**Ethical Review** Not necessary
**License** Creative Commons Attribution-NonCommercial-ShareAlike 4.0 International license (CC BY-NC-SA 4.0) https://creativecommons.org/licenses/by-nc-sa/4.0/
**First release** 2024

---

Cleaning and Labeling

---

**Alignment** All LR patches are aligned with their HR patch
**ROI** patches are cropped in a way to have at least 20% of cell content

---

Data size

---

**All tiles** 1.8 GB
**All patches** 3.4 GB

Figure 9: A data card styled (nutrition labels) for SR-CACO-2 dataset following [4, 25].

| Tile name | Size ($h \times w$) | Cell area / background area | | |
|---|---|---|---|---|
| | | CELL0 | CELL1 | CELL2 |

| | | Train set | | |
|---|---|---|---|---|
| HighRes1024-1.tif | 9318x9318 | 9.01/90.98 | 12.63/87.36 | 13.31/86.68 |
| HighRes1024-2.tif | 9318x9318 | 18.34/81.65 | 28.54/71.45 | 26.24/73.75 |
| HighRes1024-3.tif | 9318x9318 | 9.50/90.49 | 13.77/86.22 | 14.80/85.19 |
| HighRes1024-4.tif | 9318x9318 | 7.32/92.67 | 10.47/89.52 | 10.69/89.30 |
| HighRes1024-5.tif | 9318x9318 | 15.88/84.11 | 20.92/79.07 | 22.06/77.93 |
| HighRes1024-6.tif | 9317x9317 | 14.29/85.70 | 17.02/82.97 | 14.33/85.66 |
| HighRes1024-8.tif | 9317x9317 | 22.89/77.10 | 28.19/71.80 | 25.12/74.87 |
| HighRes1024-12.tif | 9318x9318 | 13.57/86.42 | 27.64/72.35 | 39.15/60.84 |
| HighRes1024-13.tif | 9318x9319 | 14.74/85.25 | 19.30/80.69 | 27.20/72.79 |
| HighRes1024-15.tif | 9319x9318 | 12.70/87.29 | 25.61/74.38 | 35.14/64.85 |
| HighRes1024-16.tif | 9318x9319 | 20.77/79.22 | 38.70/61.29 | 47.23/52.76 |
| HighRes1024-17.tif | 9318x9319 | 20.77/79.22 | 38.70/61.29 | 47.23/52.76 |
| HighRes1024-18.tif | 9319x9318 | 12.70/87.29 | 25.61/74.38 | 35.14/64.85 |
| HighRes1024-21.tif | 9318x9318 | 13.57/86.42 | 27.64/72.35 | 39.15/60.84 |
| HighRes1024-22.tif | 9317x9318 | 15.47/84.52 | 23.85/76.14 | 27.37/72.62 |
| **Average set** | – | 14.76/85.22 | 23.90/76.08 | 28.27/71.71 |

| | | Validation set | | |
|---|---|---|---|---|
| HighRes1024-7.tif | 9317x9317 | 19.90/80.09 | 25.91/74.08 | 21.27/78.72 |
| HighRes1024-11.tif | 9317x9317 | 15.77/84.22 | 19.53/80.46 | 17.84/82.15 |
| HighRes1024-19.tif | 9317x9318 | 10.16/89.83 | 17.68/82.31 | 22.65/77.34 |
| **Average set** | – | 15.27/84.71 | 21.04/78.95 | 20.58/79.40 |

| | | Test set | | |
|---|---|---|---|---|
| HighRes1024-9.tif | 9317x9317 | 11.79/88.20 | 14.76/85.23 | 11.26/88.73 |
| HighRes1024-10.tif | 9317x9317 | 19.85/80.14 | 22.67/77.32 | 19.37/80.62 |
| HighRes1024-14.tif | 9317x9318 | 10.16/89.83 | 17.68/82.31 | 22.65/77.34 |
| HighRes1024-20.tif | 9318x9319 | 14.74/85.25 | 19.30/80.69 | 27.20/72.79 |
| **Average set** | – | 14.13/85.85 | 18.60/81.38 | 20.12/79.87 |

| | | | | |
|---|---|---|---|---|
| **Total average** | – | 14.72/85.26 | 22.55/77.43 | 25.74/74.24 |

Table 3: Statistics of SR-CACO-2 dataset, HR tiles: tiles size, and foreground (cell) / background areas (%) of **high-resolution tiles**. The splits of train, validation, and test sets are done randomly. Foreground regions are determined automatically by thresholding where the threshold is set to 4 to capture all relevant parts.

where ID $\in$ ['1', '2', '3', '4', '5', '6', '7', '8', '9', '10', '11', '12', '13', '14', '15', '16', '17', '18', '19', '20', '21', '22']. Note that a tile contains all the three markers, one in each plane: plane 0 for `CELL0`, plane 1 for `CELL1`, plane 2 for `CELL2`. The names of the patches follows this format:

1. HR tiles: `HighRes1024/HighRes1024-<ID>_S_h0_h1_w0_w1_CELL.tif`

2. LR tiles (/2): `LowRes512/LowRes512-<ID>_S_h0_h1_w0_w1_CELL.tif`

3. LR tiles (/4): `LowRes256/LowRes256-<ID>_S_h0_h1_w0_w1_CELL.tif`

4. LR tiles (/8): `LowRes128/LowRes128-<ID>_S_h0_h1_w0_w1_CELL.tif`

where `S` is the patch ID, a sequential counter of the valid patches in a tile, `h0_h1_w0_w1` are the coordinates of the patch, `I` is the cell $\in$ [0, 1, 2].

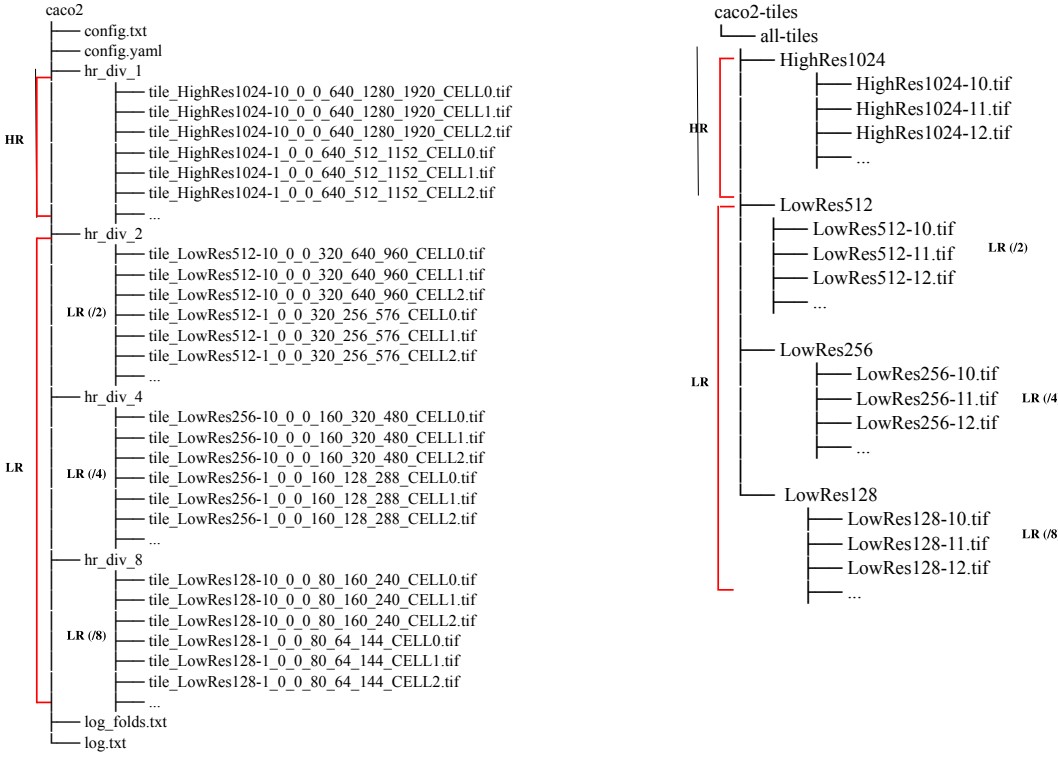

Patches                                           Tiles

Figure 10: File structure of `SR-CACO-2` dataset for patches and tiles folders.

# E   Dataset Publishing and Usage

## E.1   Dataset Hosting and Licensing

The dataset and code are available for researchers. The dataset is released under Creative Commons Attribution-NonCommercial-ShareAlike 4.0 International license (CC BY-NC-SA 4.0) https://creativecommons.org/licenses/by-nc-sa/4.0/. It is freely accessible. The dataset represents 1.8 GB for all the tiles, and 3.4 GB for all the patches. The dataset is stored in `Google Drive` [6] intended for long time availability. The `Google Drive` account is accessible by the all the authors of the dataset for future updates, and withdrawal if needed. The `Github` code web-page is meant as the front-end site for the dataset https://github.com/sbelharbi/sr-caco-2. The `ReamMe` file contains a brief overview of the dataset, its download link, link to a permanent `arXiv` [7] pdf with the content of this

___________________

[6]https://drive.google.com
[7]https://arxiv.org

paper, link to download the pretrained weights, and installation and usage of the code. The code uses an open-source 3-Clause BSD License [1]. Due to their large size, the pretrained weights of different benchmarked methods are stored in `Hugging Face` https://huggingface.co/sbelharbi/sr-caco-2 under the same license as the code.

### E.2 Intended Uses and Ethical Considerations

The `SR-CACO-2` dataset does not require ethics approval. Caco-2 (*Ca*ncer *co*li-2) cells were isolated from a 72 year old white male in the 1970s as part of a collection of gastrointestinal cancer cell lines at the Memorial Sloan Kettering Cancer Center [46]. The use of Caco-2 cells does not require ethics approval because the cells were established prior to the US Federal Policy for the Protection of Human Subjects (1991) and the UK Human Tissue Act (2004). This is the standard for biological field to use of cell lines from human that are now considered publicly available. The cells were obtained from the American Type Culture Collection (ATCC)[8], a nonprofit organization that collects and distributes cell lines and other biological materials that are publicly accessible.

The dataset is made available freely available under Creative Commons license CC BY-NC-SA 4.0. No personal information of human subject who provided the cells is available. Based on our dataset, it is impossible to uncover their identity.

The dataset is meant primarily to train models for SISR task for confocal fluorescence microscopy imaging for 3 different scales (`X2`, `X4`, and `X8`). Accurate models trained with our dataset can be efficiently used to perform SISR on live-imaging of low-resolution videos. This allows fast imaging at low-resolution, and therefore, reduce the cell damage caused by long exposure to light, and also allowing the observation of instantaneous inter-cellular events. Additionally, improving live-imaging videos quality has a huge benefit for downstream biological tasks including cell segmentation, cell counting, cell movement tracking, identifying and characterizing cell divisions.

Provided the adequate annotation, this dataset could be used for other biological/cell related applications for computational training such as: **Cell segmentation**: the availability of both nuclei and cell membranes makes cell segmentation possible. However, the dataset does not exhibit multiple different cell types/cell shapes, making it less universal. **Cell count**: Models can be trained to identify cell number, cell shape, perhaps also identifying dividing cells as well, since there is staining for chromosomes as well as Tubulin/Survivin, which both mark certain stages of division.

We are not aware of any way to misuse this dataset. The authors declare that they bear all responsibility in case of any violation of rights during the collection of the data or other work, and will take appropriate action when needed, *e.g.*, by removing data with such issues.

## F   SISR Baselines

### F.1   Evaluation Metrics

To provide quantitative comparison of different methods, we consider standard super-resolution performance measures [92]. In particular, we use Peak signal-to-noise ratio (`PSNR`), structural similarity index (`SSIM`) [91], and normalized root mean square error (`NRMSE`) [65].

Let us consider $\boldsymbol{Y}$, and $\hat{\boldsymbol{Y}}$ the ground truth, and predicted high resolution images, respectively.

The `PSNR` is one of the most common metrics for image reconstruction task [92]. It is defined as the log of the ratio between the maximum pixel value and the mean squared error (MSE) between the ground truth and the prediction,

$$\text{PSNR}(\boldsymbol{Y}, \hat{\boldsymbol{Y}}) = 10 \cdot \log_{10} \left( \frac{m^2}{\frac{1}{n} \sum_{i=1}^{n} (\boldsymbol{Y}_i - \hat{\boldsymbol{Y}}_i)^2} \right) , \tag{1}$$

where $m$ is the maximum pixel value which is $m = 255$ in the case of 8-bit image representation, and $n = h \times w$ is the total number of pixels in the image with height $h$, and width $w$. Note that high `PSNR` measure indicates better predicted image. A related measure to `PSNR` is `NRMSE` which normalizes the MSE measure as follows,

---

[8]https://www.atcc.org

$$\text{NRMSE}(\boldsymbol{Y}, \hat{\boldsymbol{Y}}) = \sqrt{\frac{1}{n}\sum_{i=1}^{n}(\boldsymbol{Y}_i - \hat{\boldsymbol{Y}}_i)^2} / (\max(\boldsymbol{Y}) - \min(\boldsymbol{Y})) \ . \tag{2}$$

Note that low `NRMSE` indicate better predicted image.

Different from `PSNR` and `NRMSE` measures which compare only pixel intensities, `SSIM` considers a neighboring window around a pixel to capture more local statistics [91]. This local context is modeled via a kernel capturing specific size. In particular, `SSIM` measure combines luminance, contrast, and structure. Since this measure has more fidelity to signal quality [72, 90], it has been largely adopted in super-resolution community [92].

We denote by $\boldsymbol{y}$, and $\hat{\boldsymbol{y}}$ as local patches at the same location from $\boldsymbol{Y}$, and $\hat{\boldsymbol{Y}}$, respectively. The `SSIM` measure at this patch is referred to as `ssim`, and it measures as,

$$\texttt{ssim}(\boldsymbol{y}, \hat{\boldsymbol{y}}) = \frac{(2\mu_{\boldsymbol{y}}\mu_{\hat{\boldsymbol{y}}} + C_1) + (2\sigma_{\boldsymbol{y}\hat{\boldsymbol{y}}} + C_2)}{(\mu_{\boldsymbol{y}}^2 + \mu_{\hat{\boldsymbol{y}}}^2 + C_1)(\sigma_{\boldsymbol{y}}^2 + \sigma_{\hat{\boldsymbol{y}}}^2 + C_2)} \ , \tag{3}$$

where $\mu_{(\cdot)}$ is the intensity mean of the corresponding patch, $\mu_{(\cdot)}$ is its corresponding variance, while $\mu_{\boldsymbol{y}\hat{\boldsymbol{y}}}$ is 2 patches covariance. The `SSIM` between two images $\boldsymbol{y}$, and $\hat{\boldsymbol{y}}$ is the average `ssim` at each location,

$$\texttt{SSIM}(\boldsymbol{Y}, \hat{\boldsymbol{Y}}) = \frac{1}{n}\sum_{i=1}^{n}\texttt{ssim}(\boldsymbol{Y}[i], \hat{\boldsymbol{Y}}[i]) \ , \tag{4}$$

where $\boldsymbol{Y}[i]$ is the patch at location $i$ of image $\boldsymbol{Y}$. High `SSIM` measure indicates better predicted image.

The evaluation is performed on the full patch, referred to as *full image*, as commonly done. We furthermore report refined performance over the cells only, referred to as *ROI only*, to better assess the prediction quality without the black background inside the patches. ROIs are determined by thresholding the HR patch using a set of thresholds[9]. A performance is computed per-threshold over the ROI only. The final reported performance is the average of all per-threshold performances. Models are trained on each cell type, and each scale, separately.

### F.2   Implementation Details

Training deep super-resolution methods typically requires defining a set of hyper-parameters. Most relevant ones are the batch size, patch size, and training loss. Before training all methods, we conduct an initial ablation over these hyper-parameters over SRCNN method [19] for `CELL2` with scale `X2`. In term of training loss, we compared three standard losses commonly used in super-resolution task [92], $\mathcal{L}_1$, $\mathcal{L}_2$, and $\mathcal{L}_{\texttt{ssim}}$. Over a single predicted image $\hat{\boldsymbol{Y}}$, and its corresponding HR ground truth $\boldsymbol{Y}$, these losses are defined as,

$$\mathcal{L}_1(\boldsymbol{Y}, \hat{\boldsymbol{Y}}) = \|\boldsymbol{Y} - \hat{\boldsymbol{Y}}\|_1 \ . \tag{5}$$

$$\mathcal{L}_2(\boldsymbol{Y}, \hat{\boldsymbol{Y}}) = \|\boldsymbol{Y} - \hat{\boldsymbol{Y}}\|_2 \ . \tag{6}$$

$$\mathcal{L}_{\texttt{ssim}}(\boldsymbol{Y}, \hat{\boldsymbol{Y}}) = \texttt{SSIM}(\boldsymbol{Y}, \hat{\boldsymbol{Y}}) \ . \tag{7}$$

Stochastic Gradient Descent (SGD) is used for their optimization. Table 4 shows the performance when using different losses. Since the quality of SR over cells is the most important, compared to black background, we opted for the case combining $\mathcal{L}_2$ and $\mathcal{L}_{\texttt{ssim}}$ which has the best performance. Therefore, all the next reported results are obtained by minimizing this composite loss,

$$\mathcal{L}(\boldsymbol{Y}, \hat{\boldsymbol{Y}}) = \mathcal{L}_2(\boldsymbol{Y}, \hat{\boldsymbol{Y}}) - \lambda\mathcal{L}_{\texttt{ssim}}(\boldsymbol{Y}, \hat{\boldsymbol{Y}}) \ . \tag{8}$$

Using Eq.8 for training, we explored other hyper-parameters. Results are reported in Fig.11. In all our next experiments, we set patch size to $96 \times 96$, batch size to 8, kernel size for `SSIM` loss to $19 \times 19$, and its $\lambda = 5$. The remaining important hyper-parameter which is the learning rate is tuned

---

[9]Evaluation thresholds are [4, 5, 6, 7, 8, 9, 10].

| Train loss / Performance | ROI only | | | Full image | | |
|---|---|---|---|---|---|---|
| | PSNR ↑ | NRMSE ↓ | SSIM ↑ | PSNR ↑ | NRMSE ↓ | SSIM ↑ |
| Bicubic | 30.38 | 0.0723 | 0.6891 | 36.33 | 0.0373 | 0.8858 |
| $\mathcal{L}_2$ | 33.13 | 0.0520 | 0.8140 | 37.23 | 0.0332 | 0.8042 |
| $\mathcal{L}_1$ | 33.03 | 0.0530 | 0.8166 | 39.17 | 0.0263 | 0.9352 |
| $\mathcal{L}_{\text{ssim}}$ | 33.19 | 0.0518 | **0.8223** | **39.23** | **0.0260** | **0.9356** |
| $\mathcal{L}_2 + \lambda\mathcal{L}_{\text{ssim}}$ | **33.42** | **0.0499** | 0.8210 | 37.81 | 0.0311 | 0.8437 |
| $\mathcal{L}_1 + \lambda\mathcal{L}_{\text{ssim}}$ | 33.38 | 0.0501 | 0.8206 | 37.75 | 0.0313 | 0.8426 |

Table 4: Ablation of training loss ablation over SR-CACO-2: performance on test set for CELL2 with scale X2 for both cases ROI only and full image; using SRCNN method [19]. Training is performed for 20 epochs.

with respect to each method, scale, and cell type over 26 values that ranges from 0.0009 to 0.01. We use 100 epochs for training amounting to a total of $\sim 92k$ SGD updates on a single NVIDIA A100 GPU. The training time of each method varies from 1.5 hours to 16 hours depending on the scale. In total, we conducted around $\sim 4.5k$ experiments with total computation time of $\sim 5k$ hours. The training time per method is presented in Tab.5.

| Methods / Scale | Training time (h) (100 epochs) | | |
|---|---|---|---|
| | X2 | X4 | X8 |
| **Pre-upsampling SR** | | | |
| SRCNN [19] *(eccv,2014)* | 3.4 | 2.0 | 1.4 |
| VDSR [42] *(cvpr,2016)* | 3.5 | 2.0 | 1.4 |
| DRRN [77] *(cvpr,2017)* | 3.6 | 2.0 | 1.3 |
| MemNet [78] *(iccv,2017)* | 11.6 | 11.6 | 11.8 |
| **Post-upsampling SR** | | | |
| NLSN [59] *(cvpr,2021)* | 5.5 | 2.5 | 2.0 |
| DFCAN [65] *(nat. methods,2021)* | 3.5 | 2.0 | 1.7 |
| SwinIR [55] *(iccvw,2021)* | 6.4 | 3.3 | 2.9 |
| EDSR (LIIF) [16] *(cvpr,2021)* | 3.4 | 2.8 | 3.0 |
| ENLCN [95] *(aaai,2022)* | 4.0 | 2.2 | 2.0 |
| GRL [52] *(cvpr,2023)* | 16.2 | 12.6 | 12.2 |
| ACT [99] *(cvpr,2023)* | 5.8 | 4.7 | 4.4 |
| Omni-SR [85] *(cvpr,2023)* | 10.5 | 9.8 | 9.9 |
| **Iterative up-and-down sampling SR** | | | |
| DBPN [28] *(cvpr,2018)* | 3.8 | 2.9 | 3.0 |
| SRFBN [54] *(cvpr,2019)* | 3.7 | 2.2 | 2.2 |
| **Progressive upsampling SR** | | | |
| ProSR [89] *(cvprw,2018)* | 3.7 | 3.5 | 3.3 |
| MS-LapSRN [44] *(tpami,2019)* | 3.4 | 2.0 | 1.3 |

Table 5: Models training time for 100 epochs.

## F.3 More Results

**SISR task.** Tables 6, 7 show more results over all the methods **on ROI only, *i.e.*, cells**, and **on full image**, respectively. Figure 12 shows the PSNR performance in a visual form across all the cells, scales, and methods. Note that performance over full images are relatively way higher compared to when only ROI is considered.

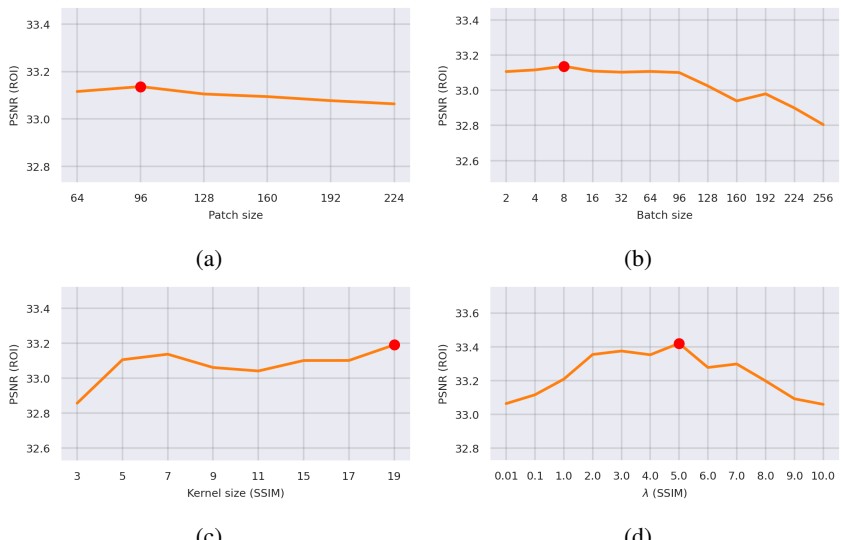

Figure 11: Ablations over general hyper-parameters: Patch size (a), batch size (b), SSIM kernel size (c), and its $\lambda$ weight (d). The PSNR performance on test set for CELL2 with scale X2 for ROI only is reported using SRCNN model [19] trained for 20 epochs.

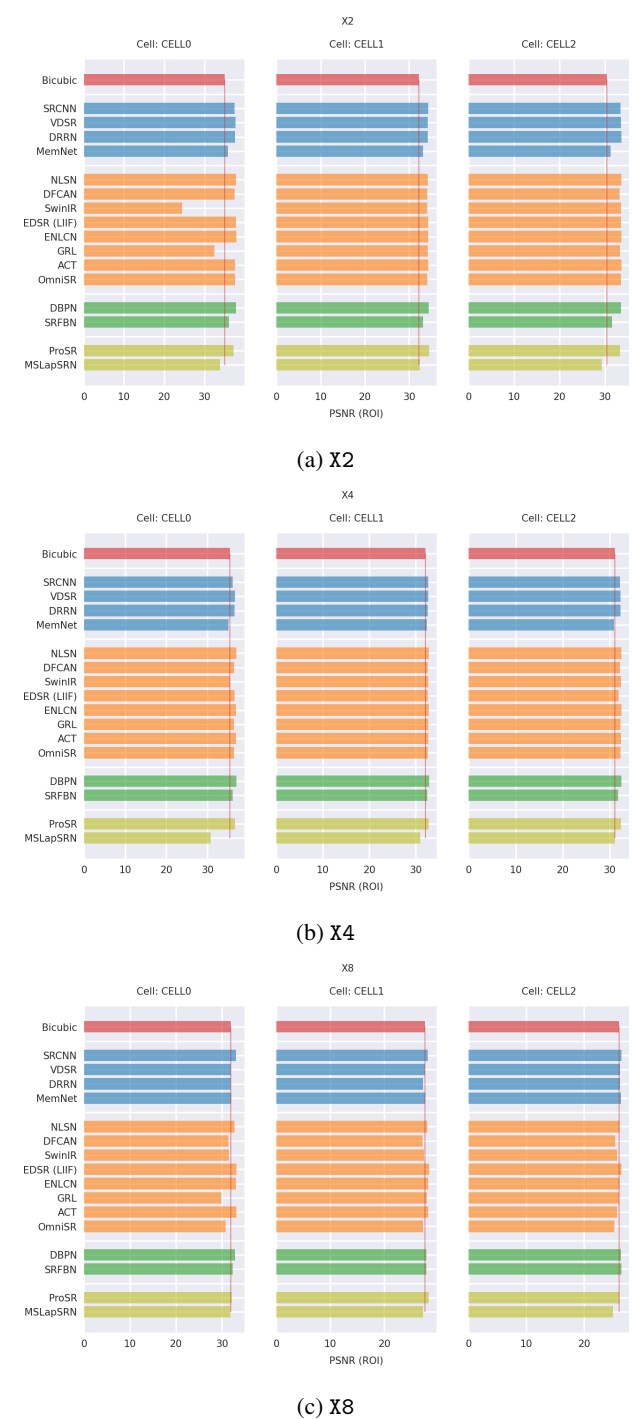

(a) X2

(b) X4

(c) X8

Figure 12: Super-resolution performance of different methods: PSNR performance over SR-CACO-2 test set **on ROI only,** *i.e.* **cells,** for the scales: X2, X4, and X8.

| SISR Methods | Scale | PSNR ↑ | | | | NRMSE ↓ | | | | SSIM ↑ | | | |
|---|---|---|---|---|---|---|---|---|---|---|---|---|---|
| | | CELL0 | CELL1 | CELL2 | Mean | CELL0 | CELL1 | CELL2 | Mean | CELL0 | CELL1 | CELL2 | Mean |
| Bicubic | X2 | 35.02 | 32.15 | 30.38 | 32.52 | 0.1085 | 0.0601 | 0.0724 | 0.0803 | 0.7618 | 0.7658 | 0.6891 | 0.7389 |
| | X4 | 35.46 | 32.03 | 31.10 | 32.86 | 0.0985 | 0.0586 | 0.0660 | 0.0744 | 0.8206 | 0.8002 | 0.7673 | 0.7960 |
| | X8 | 31.88 | 27.50 | 26.10 | 28.49 | 0.1655 | 0.1139 | 0.1349 | 0.1381 | 0.6683 | 0.6266 | 0.6511 | 0.6487 |
| **Pre-upsampling SR** | | | | | | | | | | | | | |
| SRCNN [19] *(eccv,2014)* | X2 | 37.54 | 34.27 | 33.42 | 35.08 | 0.0710 | 0.0450 | 0.0500 | 0.0553 | 0.8517 | 0.8524 | 0.8210 | 0.8417 |
| | X4 | 36.14 | 32.73 | 32.25 | 33.71 | 0.0817 | 0.0528 | 0.0572 | 0.0639 | 0.8522 | 0.8216 | 0.8079 | 0.8272 |
| | X8 | 33.05 | 28.04 | 26.49 | **29.19** | 0.1265 | 0.0967 | 0.1220 | **0.1151** | 0.7711 | 0.7085 | 0.7092 | **0.7296** |
| VDSR [42] *(cvpr,2016)* | X2 | 37.77 | 34.20 | 33.53 | 35.17 | 0.0784 | 0.0463 | 0.0498 | 0.0582 | 0.8760 | 0.8579 | 0.8334 | **0.8558** |
| | X4 | 36.69 | 32.64 | 32.31 | 33.88 | 0.0842 | 0.0536 | 0.0649 | 0.0649 | 0.8622 | 0.8262 | 0.8171 | **0.8352** |
| | X8 | 31.90 | 27.52 | 26.22 | 28.55 | 0.1651 | 0.1138 | 0.1350 | 0.1380 | 0.6695 | 0.6308 | 0.6582 | 0.6528 |
| DRRN [77] *(cvpr,2017)* | X2 | 37.61 | 34.14 | 33.59 | 35.11 | 0.0797 | 0.0464 | 0.0495 | 0.0585 | 0.8752 | 0.8570 | 0.8331 | 0.8551 |
| | X4 | 36.53 | 32.63 | 32.29 | 33.82 | 0.0860 | 0.0536 | 0.0570 | 0.0655 | 0.8620 | 0.8262 | 0.8160 | 0.8347 |
| | X8 | 31.85 | 27.21 | 26.21 | 28.42 | 0.1653 | 0.1133 | 0.1341 | 0.1375 | 0.6697 | 0.6201 | 0.6562 | 0.6486 |
| MemNet [78] *(iccv,2017)* | X2 | 35.86 | 33.17 | 31.27 | 33.43 | 0.0955 | 0.0502 | 0.0639 | 0.0699 | 0.8056 | 0.8224 | 0.7657 | 0.7979 |
| | X4 | 35.11 | 32.42 | 30.91 | 32.81 | 0.0992 | 0.0541 | 0.0675 | 0.0736 | 0.8270 | 0.8100 | 0.7605 | 0.7992 |
| | X8 | 32.00 | 27.57 | 26.41 | 28.66 | 0.1580 | 0.1044 | 0.1272 | 0.1299 | 0.6999 | 0.6717 | 0.6943 | 0.6886 |
| **Post-upsampling SR** | | | | | | | | | | | | | |
| NLSN [59] *(cvpr,2021)* | X2 | 37.89 | 34.19 | 33.62 | 35.24 | 0.0746 | 0.0452 | 0.0487 | 0.0562 | 0.8758 | 0.8541 | 0.8320 | 0.8540 |
| | X4 | 37.08 | 32.77 | 32.52 | **34.12** | 0.0751 | 0.0519 | 0.0550 | 0.0607 | 0.8592 | 0.8218 | 0.8123 | 0.8311 |
| | X8 | 32.75 | 27.96 | 26.18 | 28.96 | 0.1394 | 0.0993 | 0.1303 | 0.1230 | 0.7455 | 0.6945 | 0.6767 | 0.7055 |
| DFCAN [65] *(nat. methods,2021)* | X2 | 37.52 | 34.12 | 33.24 | 34.96 | 0.0792 | 0.0467 | 0.0514 | 0.0591 | 0.8741 | 0.8578 | 0.8321 | 0.8547 |
| | X4 | 36.43 | 32.62 | 32.17 | 33.74 | 0.0877 | 0.0541 | 0.0579 | 0.0665 | 0.8609 | 0.8265 | 0.8166 | 0.8347 |
| | X8 | 31.37 | 27.12 | 25.45 | 27.98 | 0.1656 | 0.1102 | 0.1363 | 0.1374 | 0.6927 | 0.6469 | 0.6621 | 0.6673 |
| SwinIR [55] *(iccvw,2021)* | X2 | 24.49 | 34.13 | 33.57 | 30.73 | 0.2484 | 0.0472 | 0.0494 | 0.1150 | 0.3745 | 0.8598 | 0.8330 | 0.6891 |
| | X4 | 36.56 | 32.62 | 31.92 | 33.70 | 0.0766 | 0.0532 | 0.0574 | 0.0624 | 0.8533 | 0.8231 | 0.8167 | 0.8310 |
| | X8 | 33.06 | 28.27 | 26.48 | 29.27 | 0.1261 | 0.0914 | 0.1205 | 0.1127 | 0.7700 | 0.7120 | 0.7089 | 0.7303 |
| EDSR (LIIF) [16] *(cvpr,2021)* | X2 | 37.87 | 34.27 | 33.49 | 35.21 | 0.0707 | 0.0453 | 0.0478 | 0.0546 | 0.8709 | 0.8587 | 0.8344 | 0.8547 |
| | X4 | 35.45 | 32.71 | 32.43 | 33.53 | 0.0769 | 0.0527 | 0.0559 | 0.0618 | 0.8575 | 0.8235 | 0.8154 | 0.8321 |
| | X8 | 31.53 | 27.31 | 25.77 | 28.20 | 0.1605 | 0.1077 | 0.1326 | 0.1336 | 0.7003 | 0.6586 | 0.6701 | 0.6763 |
| ENLCN [95] *(aaai,2022)* | X2 | 38.05 | 34.27 | 33.63 | **35.32** | 0.0695 | 0.0447 | 0.0485 | **0.0542** | 0.8703 | 0.8546 | 0.8284 | 0.8511 |
| | X4 | 36.96 | 32.77 | 32.48 | 34.07 | 0.0769 | 0.0521 | 0.0554 | 0.0615 | 0.8610 | 0.8233 | 0.8140 | 0.8327 |
| | X8 | 33.00 | 28.13 | 26.19 | 29.10 | 0.1335 | 0.0975 | 0.1303 | 0.1204 | 0.7563 | 0.6998 | 0.6760 | 0.7107 |
| GRL [52] *(cvpr,2023)* | X2 | 32.52 | 34.20 | 33.32 | 33.34 | 0.1200 | 0.0469 | 0.0512 | 0.0727 | 0.7764 | 0.8599 | 0.8280 | 0.8215 |
| | X4 | 36.42 | 32.72 | 32.32 | 33.82 | 0.0839 | 0.0518 | 0.0567 | 0.0641 | 0.8632 | 0.8178 | 0.8139 | 0.8317 |
| | X8 | 29.87 | 27.86 | 26.07 | 27.93 | 0.1662 | 0.0992 | 0.1179 | 0.1278 | 0.7056 | 0.7006 | 0.7095 | 0.7052 |
| ACT [99] *(cvpr,2023)* | X2 | 37.60 | 34.29 | 33.62 | 35.17 | 0.0765 | 0.0452 | 0.0483 | 0.0567 | 0.8727 | 0.8582 | 0.8230 | 0.8513 |
| | X4 | 36.86 | 32.73 | 32.43 | 34.01 | 0.0778 | 0.0518 | 0.0554 | 0.0617 | 0.8597 | 0.8177 | 0.8056 | 0.8277 |
| | X8 | 33.16 | 28.14 | 25.76 | 29.02 | 0.1265 | 0.0936 | 0.1235 | 0.1193 | 0.7678 | 0.7090 | 0.6800 | 0.7189 |
| Omni-SR [85] *(cvpr,2023)* | X2 | 37.70 | 34.11 | 33.51 | 35.11 | 0.0759 | 0.0461 | 0.0496 | 0.0572 | 0.8744 | 0.8539 | 0.8313 | 0.8532 |
| | X4 | 36.44 | 32.59 | 32.34 | 33.79 | 0.0849 | 0.0536 | 0.0563 | 0.0649 | 0.8592 | 0.8203 | 0.8111 | 0.8302 |
| | X8 | 30.75 | 27.16 | 25.30 | 27.74 | 0.1713 | 0.1098 | 0.1352 | 0.1387 | 0.6715 | 0.6419 | 0.6591 | 0.6575 |
| **Iterative up-and-down sampling SR** | | | | | | | | | | | | | |
| DBPN [28] *(cvpr,2018)* | X2 | 37.88 | 34.39 | 33.53 | 35.27 | 0.0710 | 0.0442 | 0.0491 | 0.0548 | 0.8699 | 0.8576 | 0.8285 | 0.8520 |
| | X4 | 36.97 | 32.86 | 32.48 | 34.10 | 0.0749 | 0.0514 | 0.0551 | **0.0605** | 0.8573 | 0.8253 | 0.8083 | 0.8303 |
| | X8 | 32.81 | 27.76 | 26.37 | 28.98 | 0.1303 | 0.0993 | 0.1227 | 0.1174 | 0.7650 | 0.6959 | 0.7060 | 0.7223 |
| SRFBN [54] *(cvpr,2019)* | X2 | 36.13 | 33.15 | 31.61 | 33.63 | 0.0955 | 0.0531 | 0.0625 | 0.0704 | 0.8078 | 0.8091 | 0.7470 | 0.7880 |
| | X4 | 36.08 | 32.52 | 31.79 | 33.46 | 0.0911 | 0.0545 | 0.0605 | 0.0687 | 0.8405 | 0.8147 | 0.7889 | 0.8147 |
| | X8 | 32.27 | 27.78 | 26.47 | 28.84 | 0.1560 | 0.1091 | 0.1278 | 0.1310 | 0.7022 | 0.6549 | 0.6904 | 0.6825 |
| **Progressive upsampling SR** | | | | | | | | | | | | | |
| ProSR [89] *(cvprw,2018)* | X2 | 37.35 | 34.48 | 33.32 | 35.05 | 0.0765 | 0.0434 | 0.0505 | 0.0568 | 0.8722 | 0.8556 | 0.8331 | 0.8536 |
| | X4 | 36.71 | 32.76 | 32.47 | 33.98 | 0.0810 | 0.0525 | 0.0557 | 0.0631 | 0.8626 | 0.8252 | 0.8157 | 0.8345 |
| | X8 | 32.07 | 28.23 | 26.20 | 28.83 | 0.1585 | 0.0965 | 0.1327 | 0.1293 | 0.6970 | 0.7073 | 0.6670 | 0.6904 |
| MS-LapSRN [44] *(tpami,2019)* | X2 | 33.88 | 32.36 | 29.34 | 31.86 | 0.1130 | 0.0535 | 0.0791 | 0.0819 | 0.7652 | 0.8164 | 0.7695 | 0.7837 |
| | X4 | 30.80 | 30.99 | 31.08 | 30.96 | 0.1192 | 0.0615 | 0.0626 | 0.0811 | 0.7885 | 0.7837 | 0.7806 | 0.7843 |
| | X8 | 31.83 | 27.14 | 25.06 | 28.01 | 0.1404 | 0.0982 | 0.1323 | 0.1236 | 0.7478 | 0.6933 | 0.6640 | 0.7017 |

Table 6: Super-resolution performance over SR-CACO-2 test set **on ROI only, *i.e.*, cells**.

| SISR Methods | Scale | PSNR ↑ | | | | NRMSE ↓ | | | | SSIM ↑ | | | |
|---|---|---|---|---|---|---|---|---|---|---|---|---|---|
| | | CELL0 | CELL1 | CELL2 | Mean | CELL0 | CELL1 | CELL2 | Mean | CELL0 | CELL1 | CELL2 | Mean |
| Bicubic | X2 | 41.29 | 38.23 | 36.34 | 38.62 | 0.0433 | 0.0300 | 0.0373 | 0.0369 | 0.9332 | 0.9105 | 0.8858 | 0.9098 |
| | X4 | 41.76 | 38.22 | 37.07 | 39.02 | 0.0383 | 0.0286 | 0.0337 | 0.0335 | 0.9470 | 0.9233 | 0.9128 | 0.9277 |
| | X8 | 37.82 | 32.78 | 31.01 | 33.87 | 0.0756 | 0.0702 | 0.0935 | 0.0798 | 0.8901 | 0.8447 | 0.8512 | 0.8620 |
| **Pre-upsampling SR** | | | | | | | | | | | | | |
| SRCNN [19] *(eccv,2014)* | X2 | 37.46 | 38.85 | 37.81 | 38.04 | 0.0678 | 0.0273 | 0.0312 | 0.0421 | 0.6860 | 0.8503 | 0.8438 | 0.7934 |
| | X4 | 37.59 | 37.39 | 36.99 | 37.33 | 0.0650 | 0.0319 | 0.0340 | 0.0436 | 0.7103 | 0.8157 | 0.8419 | 0.7893 |
| | X8 | 31.92 | 31.24 | 29.51 | 30.89 | 0.1327 | 0.0749 | 0.0991 | 0.1022 | 0.5038 | 0.6268 | 0.6480 | 0.5929 |
| VDSR [42] *(cvpr,2016)* | X2 | 44.57 | 40.65 | 39.74 | **41.65** | 0.0272 | 0.0214 | 0.0245 | **0.0244** | 0.9684 | 0.9527 | 0.9437 | **0.9550** |
| | X4 | 43.14 | 39.08 | 38.53 | **40.25** | 0.0312 | 0.0251 | 0.0279 | **0.0281** | 0.9611 | 0.9418 | 0.9381 | **0.9470** |
| | X8 | 37.80 | 32.72 | 31.01 | 33.84 | 0.0760 | 0.0713 | 0.0958 | 0.0810 | 0.8897 | 0.8445 | 0.8512 | 0.8618 |
| DRRN [77] *(cvpr,2017)* | X2 | 44.50 | 40.58 | 39.76 | 41.61 | 0.0273 | 0.0215 | 0.0244 | **0.0244** | 0.9691 | 0.9520 | 0.9432 | 0.9547 |
| | X4 | 43.14 | 39.03 | 38.43 | 40.20 | 0.0310 | 0.0253 | 0.0282 | 0.0282 | 0.9623 | 0.9408 | 0.9364 | 0.9465 |
| | X8 | 38.02 | 32.80 | 31.08 | **33.97** | 0.0737 | 0.0651 | 0.0939 | 0.0776 | 0.8937 | 0.8491 | 0.8536 | **0.8655** |
| MemNet [78] *(iccv,2017)* | X2 | 41.91 | 36.24 | 36.81 | 38.32 | 0.0390 | 0.0383 | 0.0344 | 0.0372 | 0.9393 | 0.7248 | 0.8754 | 0.8465 |
| | X4 | 41.39 | 36.23 | 36.89 | 38.17 | 0.0390 | 0.0374 | 0.0345 | 0.0370 | 0.9463 | 0.7588 | 0.9096 | 0.8716 |
| | X8 | 37.10 | 32.20 | 30.30 | 33.20 | 0.0777 | 0.0670 | 0.0965 | 0.0804 | 0.8495 | 0.7681 | 0.7454 | 0.7877 |
| **Post-upsampling SR** | | | | | | | | | | | | | |
| NLSN [59] *(cvpr,2021)* | X2 | 42.99 | 39.68 | 39.01 | 40.56 | 0.0333 | 0.0242 | 0.0268 | **0.0281** | 0.9164 | 0.9064 | 0.9085 | 0.9104 |
| | X4 | 39.59 | 38.12 | 37.40 | 38.37 | 0.0505 | 0.0286 | 0.0324 | 0.0372 | 0.7986 | 0.8816 | 0.8756 | 0.8519 |
| | X8 | 34.97 | 31.70 | 30.81 | 32.49 | 0.0946 | 0.0719 | 0.0916 | 0.0860 | 0.6763 | 0.6788 | 0.8225 | 0.7259 |
| DFCAN [65] *(nat. methods,2021)* | X2 | 44.27 | 40.63 | 39.49 | 41.46 | 0.0279 | 0.0214 | 0.0251 | 0.0248 | 0.9672 | 0.9533 | 0.9424 | 0.9543 |
| | X4 | 43.19 | 39.12 | 38.26 | 40.19 | 0.0307 | 0.0250 | 0.0287 | **0.0281** | 0.9641 | 0.9422 | 0.9194 | 0.9419 |
| | X8 | 37.13 | 32.51 | 30.46 | 33.36 | 0.0757 | 0.0638 | 0.0899 | **0.0764** | 0.8639 | 0.8320 | 0.8348 | 0.8435 |
| SwinIR [55] *(iccvw,2021)* | X2 | 28.93 | 39.81 | 39.67 | 36.14 | 0.1555 | 0.0228 | 0.0247 | 0.0677 | 0.3802 | 0.9228 | 0.9429 | 0.7487 |
| | X4 | 42.00 | 38.73 | 38.23 | 39.65 | 0.0347 | 0.0263 | 0.0291 | 0.0300 | 0.9257 | 0.9238 | 0.9266 | 0.9253 |
| | X8 | 36.85 | 32.35 | 30.60 | 33.27 | 0.0768 | 0.0656 | 0.0896 | 0.0774 | 0.8482 | 0.8026 | 0.8297 | 0.8268 |
| EDSR (LIIF) [16] *(cvpr,2021)* | X2 | 38.46 | 38.36 | 38.38 | 38.40 | 0.0604 | 0.0285 | 0.0276 | 0.0388 | 0.7177 | 0.8308 | 0.8908 | 0.8131 |
| | X4 | 37.48 | 36.80 | 37.81 | 37.36 | 0.0656 | 0.0335 | 0.0289 | 0.0427 | 0.7034 | 0.8008 | 0.9323 | 0.8122 |
| | X8 | 31.92 | 29.28 | 28.66 | 29.95 | 0.1327 | 0.0944 | 0.1070 | 0.1114 | 0.5033 | 0.4892 | 0.5820 | 0.5248 |
| ENLCN [95] *(aaai,2022)* | X2 | 40.20 | 39.53 | 38.62 | 39.45 | 0.0482 | 0.0248 | 0.0283 | 0.0338 | 0.8010 | 0.8939 | 0.8958 | 0.8636 |
| | X4 | 40.03 | 38.33 | 37.74 | 38.70 | 0.0476 | 0.0278 | 0.0309 | 0.0354 | 0.8221 | 0.8929 | 0.8946 | 0.8699 |
| | X8 | 33.93 | 31.30 | 30.83 | 32.02 | 0.1060 | 0.0767 | 0.0916 | 0.0914 | 0.6086 | 0.6401 | 0.8267 | 0.6918 |
| GRL [52] *(cvpr,2023)* | X2 | 28.99 | 39.79 | 39.56 | 36.11 | 0.1827 | 0.0230 | 0.0250 | 0.0769 | 0.3964 | 0.9167 | 0.9377 | 0.7503 |
| | X4 | 41.74 | 36.18 | 37.60 | 38.50 | 0.0370 | 0.0374 | 0.0311 | 0.0351 | 0.9115 | 0.7372 | 0.8681 | 0.8389 |
| | X8 | 27.93 | 31.11 | 29.12 | 29.39 | 0.2013 | 0.0767 | 0.0923 | 0.1235 | 0.3765 | 0.6283 | 0.6294 | 0.5447 |
| ACT [99] *(cvpr,2023)* | X2 | 40.92 | 38.26 | 36.36 | 38.51 | 0.0443 | 0.0289 | 0.0379 | 0.0370 | 0.8300 | 0.8243 | 0.7641 | 0.8061 |
| | X4 | 39.07 | 36.76 | 35.36 | 37.07 | 0.0542 | 0.0345 | 0.0420 | 0.0436 | 0.7717 | 0.7813 | 0.7411 | 0.7647 |
| | X8 | 32.27 | 29.80 | 27.62 | 29.90 | 0.1281 | 0.0891 | 0.1225 | 0.1133 | 0.5190 | 0.5196 | 0.5405 | 0.5264 |
| Omni-SR [85] *(cvpr,2023)* | X2 | 43.51 | 40.35 | 39.51 | 41.12 | 0.0308 | 0.0222 | 0.0252 | 0.0261 | 0.9516 | 0.9486 | 0.9412 | 0.9471 |
| | X4 | 42.14 | 38.63 | 37.94 | 39.57 | 0.0352 | 0.0265 | 0.0301 | 0.0306 | 0.9488 | 0.9313 | 0.9209 | 0.9337 |
| | X8 | 36.05 | 32.08 | 29.97 | 32.70 | 0.0815 | 0.0673 | 0.0904 | 0.0797 | 0.8586 | 0.7946 | 0.8022 | 0.8185 |
| **Iterative up-and-down sampling SR** | | | | | | | | | | | | | |
| DBPN [28] *(cvpr,2018)* | X2 | 40.15 | 39.50 | 38.19 | 39.28 | 0.0485 | 0.0249 | 0.0294 | 0.0343 | 0.8005 | 0.8859 | 0.8575 | 0.8480 |
| | X4 | 37.87 | 38.19 | 35.96 | 37.34 | 0.0634 | 0.0284 | 0.0386 | 0.0435 | 0.7108 | 0.8772 | 0.7735 | 0.7871 |
| | X8 | 32.85 | 31.34 | 29.24 | 31.14 | 0.1186 | 0.0733 | 0.1015 | 0.0978 | 0.5482 | 0.6526 | 0.6334 | 0.6114 |
| SRFBN [54] *(cvpr,2019)* | X2 | 42.41 | 39.20 | 37.41 | 39.67 | 0.0373 | 0.0265 | 0.0329 | 0.0322 | 0.9461 | 0.9258 | 0.9011 | 0.9243 |
| | X4 | 42.24 | 38.20 | 37.07 | 39.17 | 0.0358 | 0.0286 | 0.0339 | 0.0328 | 0.9498 | 0.8993 | 0.8802 | 0.9098 |
| | X8 | 37.26 | 32.72 | 30.53 | 33.50 | 0.0777 | 0.0698 | 0.0958 | 0.0811 | 0.8523 | 0.8231 | 0.7665 | 0.8140 |
| **Progressive upsampling SR** | | | | | | | | | | | | | |
| ProSR [89] *(cvprw,2018)* | X2 | 42.07 | 38.68 | 39.17 | 39.97 | 0.0368 | 0.0280 | 0.0261 | 0.0303 | 0.8996 | 0.8356 | 0.9325 | 0.8892 |
| | X4 | 41.65 | 38.61 | 38.08 | 39.45 | 0.0376 | 0.0267 | 0.0295 | 0.0313 | 0.9050 | 0.9044 | 0.9095 | 0.9063 |
| | X8 | 37.10 | 31.20 | 30.91 | 33.07 | 0.0790 | 0.0774 | 0.0937 | 0.0834 | 0.8511 | 0.6189 | 0.8376 | 0.7692 |
| MS-LapSRN [44] *(tpami,2019)* | X2 | 29.89 | 34.03 | 32.47 | 32.13 | 0.1672 | 0.0499 | 0.0572 | 0.0915 | 0.4078 | 0.6260 | 0.6772 | 0.5703 |
| | X4 | 30.99 | 31.81 | 33.69 | 32.16 | 0.1277 | 0.0657 | 0.0493 | 0.0809 | 0.4628 | 0.5500 | 0.6905 | 0.5678 |
| | X8 | 28.31 | 27.61 | 25.46 | 27.13 | 0.1995 | 0.1089 | 0.1402 | 0.1496 | 0.3803 | 0.4243 | 0.4176 | 0.4074 |

Table 7: Super-resolution performance over `SR-CACO-2` test set **on full image**.

**Downstream biology tasks: Object detection/segmentation.** Figures 13, 14 show more results for cell detection across the 2 extreme cases: CELL2 and CELL0, respectively. Cell segmentation performance is reported in Fig.15. Overall, several methods (orange) achieved comparable results to HR images on these tasks which is promising. Figure 16 shows an example of cell segmentation of different methods.

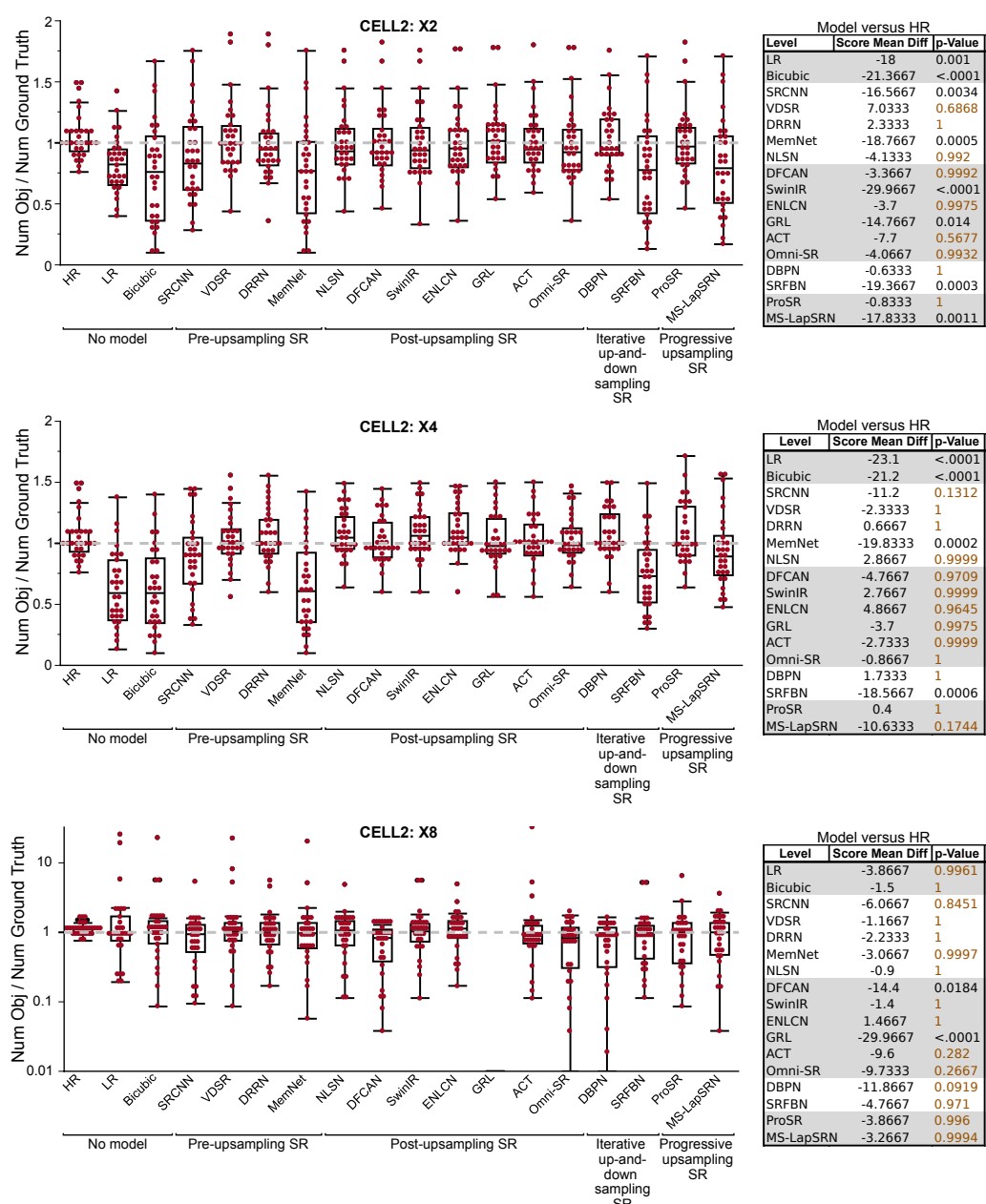

Figure 13: Analysis of cell detection performance: CELL2.

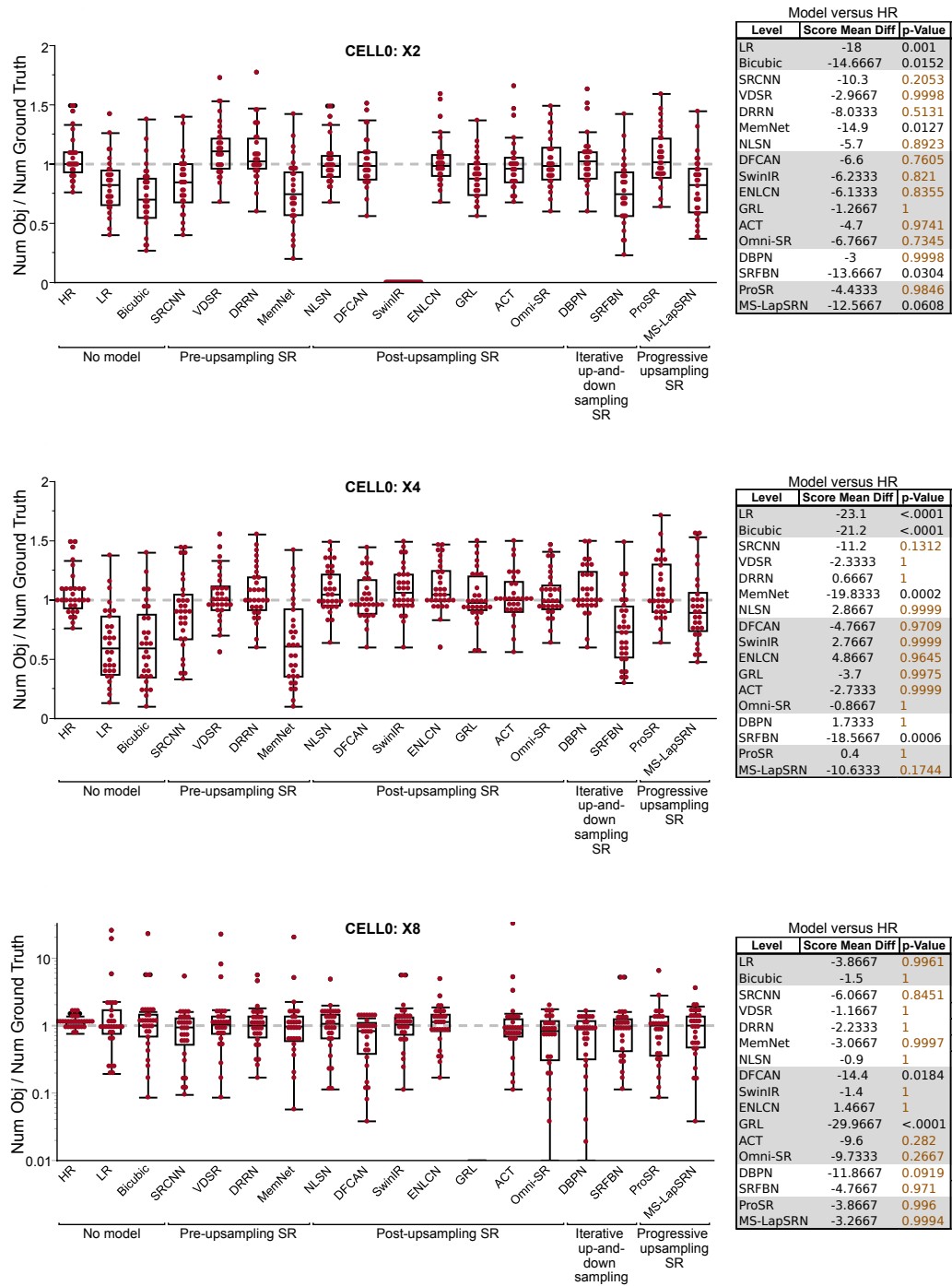

Figure 14: Analysis of cell detection performance: CELL0.

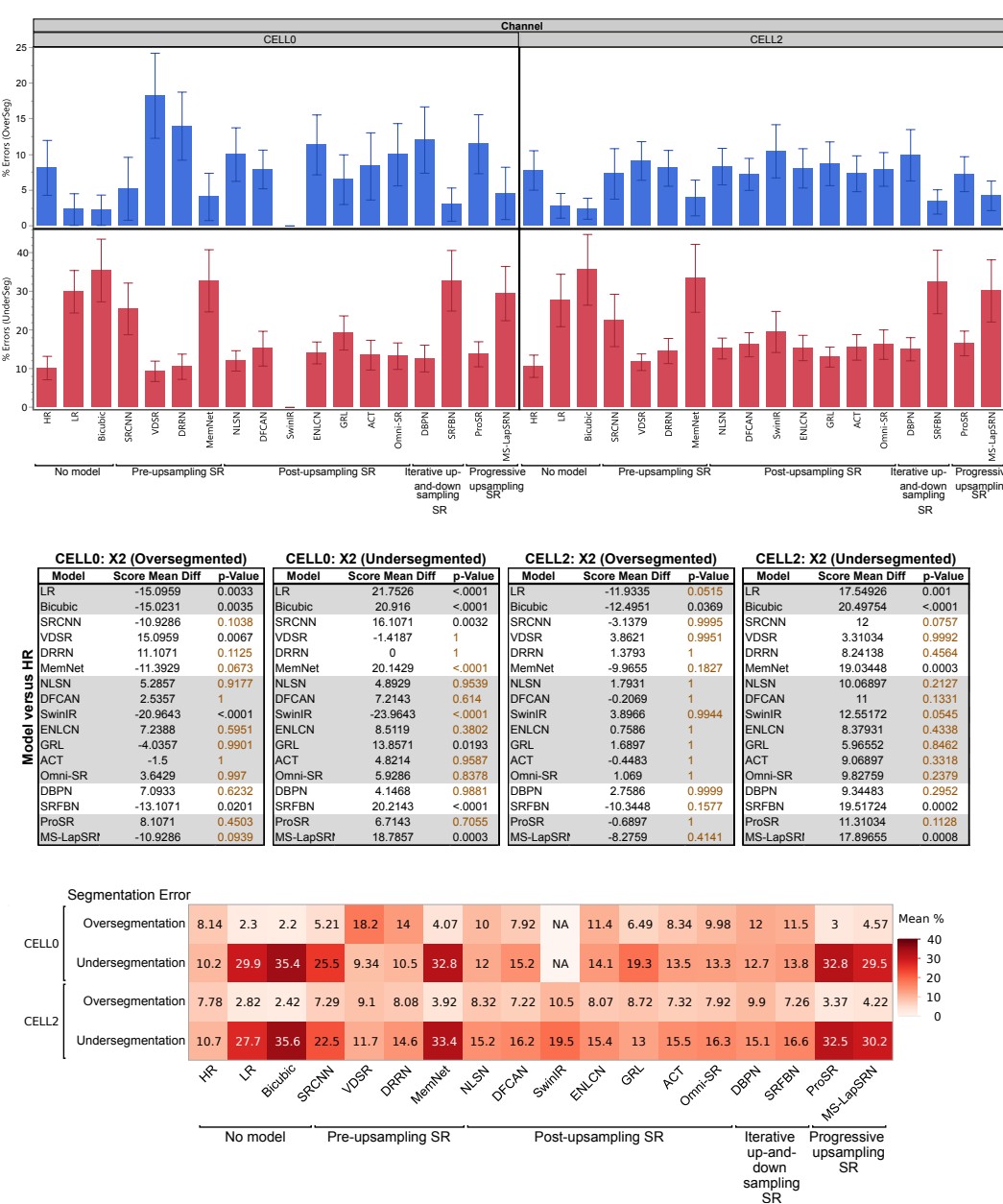

Figure 15: Analysis of cell segmentation performance: CELL0, CELL2, X2.

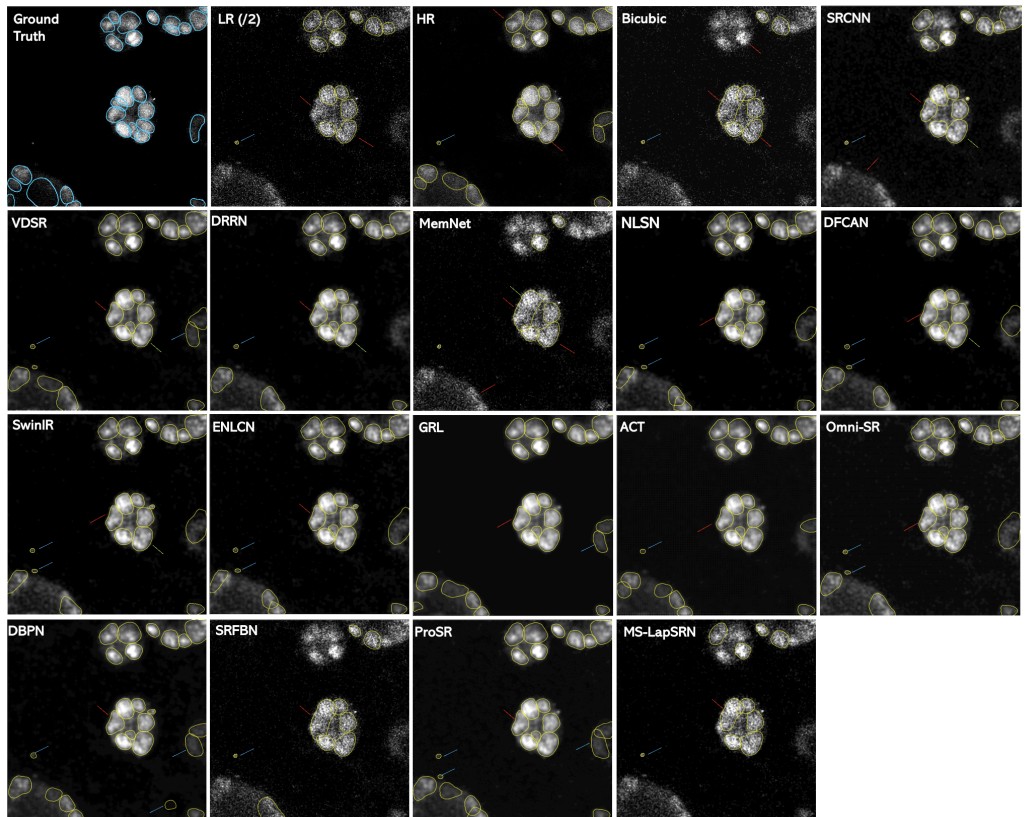

Figure 16: Cell segmentation example using different methods: CELL2, X2. Red arrow for undersegmented errors; Blue arrow for oversegmented error; and green arrow for boundary error. HR patch file sample name: hr_div_1/tile_HighRes1024-20_396_6912_7552_5632_6272_CELL2.tif. In all cases, the brightness has been enhanced just for visualization. Best visualized in color.

