# OpenReview forum: "SR-CACO-2: A Dataset for Confocal Fluorescence Microscopy Image Super-Resolution"
_NeurIPS.cc/2024/Datasets_and_Benchmarks_Track — NeurIPS 2024 Track Datasets and Benchmarks Poster_

### Official Review · Reviewer_zpaL · 2024-07-14
**put author names in ths submission**

**Rating:** 3
**Confidence:** 4
**Correctness:** The dataset is designed in a right way.
**Clarity:** Clarity is fine.

**Review:**

(1)	The primary strength lies in the construction of a dataset specifically designed for confocal fluorescence microscopy SISR, as well as the evaluation of performance on confocal fluorescence microscopy images.
(2)	The application of super resolution to these images shows promising results in comparison to both low-resolution (LR) and high-resolution (HR) counterparts, as demonstrated through downstream biology tasks such as cell object (nucleus) detection and segmentation.

**Strengths:**

(1)	The primary strength lies in the construction of a dataset specifically designed for confocal fluorescence microscopy SISR, as well as the evaluation of performance on confocal fluorescence microscopy images.
(2)	The application of super resolution to these images shows promising results in comparison to both low-resolution (LR) and high-resolution (HR) counterparts, as demonstrated through downstream biology tasks such as cell object (nucleus) detection and segmentation.

**Additional Feedback:**

put author names in this submission, Violates the principle of anonymity

**Documentation:**

The dataset and baselines are well-documented and available. However, some details of the dataset are unclear, such as diversity and statistical information.

**Ethics:**

I do not suspect any ethical concerns.

**Limitations:**

The article lacks a sufficient description and experimental comparison of related work specifically focused on real-world SISR.

**Opportunities For Improvement:**

(1)	The motivation behind this work can be enhanced by considering the challenges of real-world super resolution. Due to the significant degradation gap between synthetic kernels (such as bicubic) and real-world degradations, the models trained on natural datasets are not applicable to confocal fluorescence microscopy images. Furthermore, the comparison methods used in the evaluation of the dataset also lack real-world SISR methods for comprehensive assessment.
(2)	Some details of the dataset, such as its diversity and statistical information, remain unclear. To enhance the article, you may refer to the RealVSR[1] dataset, which provides valuable insights for improvement.
(3)	The article primarily focuses on presenting a dataset for evaluation purposes, but it could benefit from further exploration and discussion regarding the potential impact of this dataset on downstream tasks. It would be valuable to consider how this dataset can contribute to and enhance various downstream tasks.
(4)	It is important to mention and compare this work with similar studies, such as the work presented in [3], which also addresses microscopy SISR and utilizes a constructed dataset.
[1]. Yang, Xi, et al. "Real-world video super-resolution: A benchmark dataset and a decomposition based learning scheme." Proceedings of the IEEE/CVF International Conference on Computer Vision. 2021.
[2]. Chen, Honggang, et al. "Real-world single image super-resolution: A brief review." Information Fusion 79 (2022): 124-145.
[3].  Ma, Jiabo, et al. "PathSRGAN: multi-supervised super-resolution for cytopathological images using generative adversarial network." IEEE transactions on medical imaging 39.9 (2020): 2920-2930.

**Relation To Prior Work:**

It clearly discusses how the work differs from previous SISR setting, but real-world SISR methods are not included.

**Summary And Contributions:**

In this work, the first-ever large and challenging dataset, SR-CACO-2, specifically designed for confocal fluorescence SISR microscopy is presented. The dataset consists of 9937 patches and includes three corresponding real LR versions (/2, /4, and /8) covering three distinct protein markers. A comprehensive benchmarking study evaluating the quality of super-resolved images is conducted, involving 15 representative SISR methods.

---

> ### Author Rebuttal · Authors · 2024-08-17
>
> Thank you for your feedback. We now address your comments under "Opportunities For Improvement", "Limitations", and "Additional Feedback":
>
> - **Question: "Can you include real-world SISR methods?"**
>
> Reply:
>
> Thank you for this suggestion. Yes, we will include a discussion about this family of methods. We note that the current literature is dominated by SISR methods that use interpolation as a way to construct LR images. In addition, recently published works on microscopy super-resolution such as [5, 32] perform a standard comparison to SISR methods. Moreover, initial results obtained with realIS are not a breakthrough compared to SISR methods (almost +1% gain) [REF4]. Nonetheless, real-world SISR methods have recently emerged, and it makes sense to describe and compare these methods. Depending on the availability of the public code, we aim to include results on selected realIS methods by the camera-ready deadline.
>
> A kind reminder that the [NeurIPS Datasets and Benchmarks Track](https://neurips.cc/Conferences/2024/CallForDatasetsBenchmarks) allows for the publication of works that propose new datasets and/or benchmarking studies. Our work aims primarily to introduce the new SR-CACO-2 dataset for confocal super-resolution. A secondary contribution consists of benchmarking several representative methods selected across different SISR families to assess their performance and limitations. This benchmarking study can provide a reference point for comparison with future works. Therefore, we believe that not comparing to a SISR family should not be used as a reason to reject the paper.
>
> - **Question: "Some details of the dataset, such as its diversity and statistical information, remain unclear."**
>
> Reply:
>
> We have provided relevant statistics about our datasets in the main paper and the supplementary materials. As for the diversity, it is difficult to quantify. However, our dataset exhibits different and relevant factors of diversity to make it valuable and real-world data for confocal microscopic super-resolution fields. Please see our more detailed answer to this question for reviewer **kHKG**, [here](https://openreview.net/forum?id=GtYd9PCaaB&noteId=MrSX0AqODq) and [here](https://openreview.net/forum?id=GtYd9PCaaB&noteId=7pZ2UcuHw3).
>
> In addition, we would like to clarify that the statistics presented in Tab.1 of the main paper are per cell and scale. This means that each cell and scale have $\sim$ 10k patches. We provide 3 cell types and 4 scales.
>
> It would be also helpful if you could clarify the exact statistics we are missing so that we can include them.
>
> - **Question: "Potential impact of this dataset on downstream tasks."**
>
> Reply:
>
> We have already presented a discussion and results with quantified performance on two downstream tasks: cell detection and segmentation in the main paper and supplementary material. Our results indicate that some SISR methods can yield very competitive performance on these tasks compared to HR images. We are currently working to collect additional data to further evaluate trained SISR models on our datasets on the task of tracking in video microscopy imaging. This will allow training on static images and applying them to time-lapse imaging to determine if cell detection can improve cell tracking.
>
> As mentioned in the paper, the ultimate goal of using this dataset is to train SISR models to produce accurate super-resolution images. Then, in practice, one can capture low-resolution live imaging under reduced light exposure to preserve cells. This leads to fast imaging which allows the observation of instantaneous inter-cellular events with less damage to the cells. Then, a model trained on our dataset can be used to upscale the images and perform downstream tasks such as cell tracking, counting, and segmentation, without the need for HR images.
>
> - **Question: "It is important to mention and compare this work with similar studies, such as the work presented in PathSRGAN."**
>
> Reply:
>
> Thank you for this suggestion. We will discuss and compare PathSRGAN in our revised manuscript.
>
> - **Question: "put author names in this submission, Violates the principle of anonymity."**
>
> Reply:
>
> A kind reminder that the NeurIPS Datasets and Benchmarks Track is different from the main NeurIPS conference. Its main purpose is to publish papers dedicated to new datasets and/or benchmarks. This track has its guidelines for authors and reviewers. Reviewing in this track is primarily single-blind. Only some submissions are allowed to be reviewed anonymously. An example is when authors can provide all required material by this track such as data access, code, and pretrained weights without revealing their identity while ensuring that the reviewers can perform their reviewing jobs without an issue. This done under the risk of the authors. In our case, we have to provide large files of pretrained weights via Hugging Face account. At the end, it is up to the authors. (See the guidelines for conditions: [Track](https://neurips.cc/Conferences/2024/CallForDatasetsBenchmarks).)
>
> Most of your comments relate to the comparison and discussion of other methods. While this will help improve the paper, we believe that it is not grounds for a "clear rejection". Indeed, the main purpose of our paper is to propose our new dataset and not to benchmark different super-resolution methods. We hope that our rebuttal will help clarify your questions, and justify raising your overall rating of our manuscript.

---

> > ### Author Response · Authors · 2024-08-21
> > **follow up**
> >
> > Dear Reviewer "zpaL",
> >
> > We sincerely thank you for your precious time and efforts in reviewing our paper. We believe our responses have addressed your concerns including the discussion of other methods, dataset diversity, and its impact on downstream tasks. If you have any additional questions or require further clarification, we would be happy to provide it.
> >
> > We are looking forward to your reply.
> >
> > Best -- The Authors

---

> > > ### Author Response · Authors · 2024-08-29
> > > **reminder**
> > >
> > > Dear Reviewer "zpaL",
> > >
> > > We thank you again for your effort to review our work. This is a kind reminder that the deadline for discussion is closing in less than 2.5 days (Aug. 31th).
> > > We hope that you can read our rebuttal before this deadline.
> > > We also hope that we have answered all your questions and clarify any misunderstanding to help improve your score.
> > > We will be happy to answer any other questions.
> > > We look forward to your response.
> > >
> > > Thank you again
> > >
> > > Best
> > >
> > > All authors

---

> > ### Author Response · Authors · 2024-09-01
> > **reminder**
> >
> > Dear Reviewer "zpaL",
> >
> > This is a kind reminder that the deadline for discussion is closing in less than **1 hour 50min**. We hope that you can read our rebuttal before this deadline. We also hope that we have answered all your questions and clarify any misunderstanding to help improve your score. We will be happy to answer any other questions. We look forward to your response.
> >
> > Thank you again
> >
> > Best
> >
> > All authors

---

### Official Review · Reviewer_zxUt · 2024-07-21
**The generated data can assist researchers in developing data-driven models that effectively address real-world challenges.**

**Rating:** 7
**Confidence:** 4
**Correctness:** Yes
**Clarity:** Yes

**Review:**

The generated data enables the machine learning community to test the performance of super-resolution models on real experimental data. This helps identify current challenges and areas for improvement in deep learning architectures to better tackle real-world problems.

**Strengths:**

This paper is well-written, clearly demonstrating the significance of the proposed dataset. The experimental results of the benchmark deep learning models are also convincing.

**Additional Feedback:**

The paper is recommended for acceptance.

**Documentation:**

Yes

**Limitations:**

Please consider discussing the limitations of the proposed dataset.

**Opportunities For Improvement:**

There are still some minor issues that can be improved:

1. Some experiments in Table 2 yield surprising results and require further explanation. For example, the performance of MS-LapSRN [22] (TPAMI, 2019) is worse than bicubic interpolation in terms of PSNR. Please elaborate on these experiments and explain the potential reasons behind this poor performance.

2. While the authors provided extensive experiments using SOTA deep learning architectures, it would be beneficial to compare these models with conventional iterative solvers like TV-regularization. This comparison is particularly interesting to demonstrate how deep learning can surpass traditional iterative solvers in practice using your proposed experimental data.

3. An important family of deep learning architectures for super-resolution includes methods based on neural fields, which can recover high-resolution images at any arbitrary resolution. These models are crucial for high-dimensional image reconstruction common in practice, like your generated data. The authors are recommended to add this family to your benchmark list or at least mention these methods in your paper.

[1] Chen, Yinbo, Sifei Liu, and Xiaolong Wang. "Learning continuous image representation with local implicit image function." CVPR 2021.

[2] AmirEhsan Khorashadizadeh, Anadi Chaman, Valentin Debarnot, and Ivan Dokmanic. "FunkNN: Neural interpolation for functional generation." ICLR 2023.

[3] Gao, Sicheng, et al. "Implicit diffusion models for continuous super-resolution." CVPR 2023.

**Relation To Prior Work:**

Yes

**Summary And Contributions:**

This paper presents SR-CACO-2, a large-scale confocal microscopy dataset consisting of paired low- and high-resolution images. This dataset is vital for machine learning researchers aiming to train deep neural networks for image super-resolution tasks using real-world data. The authors tested state-of-the-art deep learning models on this dataset, showing that current SOTA architectures fall short in producing high-quality, sharp reconstructions.

---

> ### Author Rebuttal · Authors · 2024-08-17
>
> Thank you for your feedback and positive comments. We are pleased that you found our work interesting and valuable for the research community. We now address your comments under "Opportunities For Improvement" and "Limitations":
>
> - **Question: "MS-LapSRN in Table 2 yields surprising results and requires further explanation."**
>
> Reply:
>
> Thank you for this observation. We have analyzed the reason for this decline in performance and found that the model failed to converge properly as the loss did not decrease as expected. This is most clear on CELL0 data leading to a large decline compared to CELL1 and CELL2. We believe that this is caused by the combination of the very low brightness level of the cell and the use of residual learning in MS-LapSRN. This model has difficulty producing very small residuals but instead, it yields large residuals. This can be confirmed when measuring performance over a full image, i.e., by including a dark background where the typical intensity is 0 which leads to a sharp decline in performance.
> Refer to Tab.5 (supplementary materials), where LapSRN yields a PSNR of 29.89 for X2, vs 41.29 for bicubic. By inspecting some predicted samples, the intensity at these background regions is typically 10. Note that this is not the case for the ProSR model that also uses residual learning. However, ProSR employs a different architecture than MS-LapSRN which, in this case, seems much more efficient due to its depth or its local residual layers. We will include this discussion in the paper.
>
> During our experiments, we observed that hyper-parameters do not transfer well for a model across cell types, or scales. Therefore, we performed a search over a grid separately for each model, for each cell type, and for each scale. This provides for each method with a good and fair chance to perform well. Cases of failure are mostly related to the method itself and the nature of the data.
>
>
> - **Question: "Can you include results using iterative solvers like TV-regularization, and neural field methods or at least discuss them?"**
>
> Reply:
>
> Indeed, even though we have performed a comparison of different SISR families, we certainly have missed some methods. Thank you for your suggestions. We will include a discussion about these methods in the paper and cite the three recommended papers. Implementing, and running experiments in this short time is very challenging. However, we will be able to include the results for some of the methods you suggested by the camera-ready deadline. We have studied these methods, they already have a PyTorch public implementation which will help us run our experiments.
>
> - **Question: "Please consider discussing the limitations of the proposed dataset."**
>
> Reply:
>
> In this NeurIPS Datasets and Benchmarks Track, the main paper cannot hold all the necessary details of a proposed dataset. We are required to include all the details in the supplementary materials: [Track website](https://neurips.cc/Conferences/2024/CallForDatasetsBenchmarks). We kindly invite you to verify the supplementary materials we submitted. It contains the following points, including a discussion of the limitations of the proposed dataset (Section C):
>
> A. Data Access for Review
>
> B. Related Work
>
> C. SR-CACO-2 Dataset Limitations
>
> D. Overview of Dataset
>
> D.1 Dataset statistics
>
> D.2 Data Format and Organization
>
> E. Dataset Publishing and Usage
>
> E.1 Dataset Hosting and Licensing
>
> E.2 Intended Uses and Ethical Considerations
>
> F. SISR Baselines
>
> F.1 Evaluation Metrics
>
> F.2 Implementation Details
>
> F.3 More Results
>
> We will integrate our response to your comments in our manuscript. Hopefully, our response will justify raising your overall rating of our manuscript.

---

> > ### Comment · Reviewer_zxUt · 2024-08-29
> >
> > Thanks for addressing my concerns. It seems the manuscript has not been updated yet! Please apply the promised changes and update your file.

---

> > > ### Author Response · Authors · 2024-08-29
> > > **updates on openreview.**
> > >
> > > Dear Reviewer "zxUt",
> > >
> > > Thank you for your timely reply and consideration to our rebuttal.
> > > We are glad that you found our rebuttal satisfactory.
> > > As for the updates, unfortunately, openreview does not seem to give us the option to update the submission (during the rebuttal or at the moment. please see a screenshot of the website [here](https://drive.google.com/file/d/1JFCAuS_nSJZyD5H1v082RPfU3tP_nfFv/view?usp=sharing)). Updates at this period may have been disabled for this track by the organizers. However, please rest assured that we will include your comments and the comments of other reviewers in the final version.
> > >
> > > Thank you again,
> > > Best
> > > The authors

---

### Official Review · Reviewer_wsnJ · 2024-07-23
**A large benchmark for the super resolution of Confocal Fluorescence Microscopy Image**

**Rating:** 8
**Confidence:** 4
**Correctness:** Yes
**Clarity:** Yes

**Review:**

Creating a large microscopy image dataset with various resolutions is meaningful for designing single-image-super-resolution methods, but it is also very challenging. The paper presents a large dataset of this kind, which will be very useful for further development of new methods. More interestingly, the author points out the potential issues of creating low-resolution images in previous datasets and use the low-resolution setting of a microscope to create more realistic imaging. Fifteen methods were tested using this dataset, which is extensive and useful for further studies.

**Strengths:**

(1) Use the same microscope to generate the dataset with pairs of high-resolution and low-resolution.

(2) Create a large-scale dataset

(3) Benchmarking 15 methods, which shows a clear picture of how state-of-art methods perform.

(4) Give clear instructions for the public to access the dataset.

**Additional Feedback:**

This is an exciting and beautiful work.

**Documentation:**

Yes

**Ethics:**

No, I don't.

**Limitations:**

Yes, the authors have adequately addressed them.

**Opportunities For Improvement:**

Though theoretically, interpolation can be a too simple way to generate low-resolution images, it would be really beneficial for readers to understand how different the "real" LR images and "mathematically downsampling" LR images are.

**Relation To Prior Work:**

Yes

**Summary And Contributions:**

The paper presents a large confocal microscope imaging dataset of cells in 3D with both low-resolution and high-resolution samples.  The author can capture samples with realistic low-resolution data by modifying the microscope's settings. In addition to the data, the author benchmarked 15 state-of-the-art methods, which show these methods have limited performance.

---

> ### Author Rebuttal · Authors · 2024-08-17
>
> Thank you for your feedback and positive comments. We are pleased that our work is met with your enthusiastic review. We now address your comment under "Opportunities For Improvement":
> - **Question: "It would be really beneficial for readers to understand how different the "real" LR images and "mathematically downsampling" LR images are."**
>
> Reply:
>
> Thank you for this interesting question. Unfortunately, we do not have a formulation to directly compare real low-resolution images with interpolated low-resolution images. However, an empirical analysis was undertaken to compare both images: real LR vs. bicubic LR. Here are some relevant observations:
>
> - Pixel-intensity difference: both LR types are different. Their absolute pixel-wise difference value can be in [0, 50] as well as [200, 255] (see the |REAL - BICUBIC| histogram in the provided samples). The difference is mainly concentrated over cell regions.
>
> - Compared to real LR, bicubic LR can maintain better cell structure and even full cells from the HR images (see the real and bicubic images). A good example is the image: [link-1](https://drive.google.com/file/d/1oJBsi0gZ-ItqaaOu1CxM7g1AO-VJr0uH/view?usp=sharing), [link-2](https://drive.google.com/file/d/1YwlSKBdceLIqiMrxN4fgxYW1TKvKN6wa/view?usp=sharing).
>
> - Intensity span: depending on the image, the real LR images are sometimes much more expressive in terms of pixel intensity as they span larger intensities compared to bicubic LR (see the pixel-intensity histogram).
>
> - Real LR images are more noisy compared to bicubic LR ones which tend to be very smooth (see the visual result of the Laplace filter, and the histogram of its absolute value |Laplace LR|). Note that the noise in real LR is the results of a single scan while HR are scanned 9 times to be averaged. This is a typical example: [link](https://drive.google.com/file/d/1ZaRr-XkreeAks2LxpTHNdl9YlmrKPRrN/view?usp=sharing).
>
> More examples taken from the trainset across all cells/scales can be found here: 10 examples (26MB, [link-10](https://drive.google.com/file/d/1QrsK7ucsqxxzRkH3Sy2-cZVMBRmQrPIB/view?usp=sharing)), 100 examples (267MB, [link-100](https://drive.google.com/file/d/1BE-fIMbWMKTqFJplfgf6sWJ_16kv__Sb/view?usp=sharing)).
>
> This large contrast between real and bicubic LR only confirms that bicubic LR cannot be used as a replacement to substitute real LR images for training and evaluation, as done in SR methods over natural scene images. Real LR images must be used to effectively simulate a realistic scenario for the model at deployment time. Therefore our collection of real LR images is extremely valuable to designing realistic SR models.
>
> We plan to integrate our response to your comments in our revised manuscript. Hopefully, our response will justify raising your overall rating of our manuscript.

---

> > ### Author Response · Authors · 2024-08-29
> > **reminder**
> >
> > Dear Reviewer "wsnJ",
> >
> > We thank you again for your effort to review our work. We are glad that you liked our work. This is a kind reminder that the deadline for discussion is closing in less than 2.5 days (Aug. 31th).
> > We hope that you can read our rebuttal before this deadline.
> > We also hope that we have answered your question or at least gave some insights.
> > We will be happy to answer any other questions.
> > We look forward to your response.
> >
> > Thank you again
> >
> > Best
> >
> > All authors

---

> > > ### Author Response · Authors · 2024-09-01
> > > **reminder**
> > >
> > > Dear Reviewer "wsnJ",
> > >
> > > We thank you again for your effort to review our work. This is a kind reminder that the deadline for discussion is closing in less than **1 hour 50min**. We hope that you can read our rebuttal before this deadline. We also hope that we have answered your question or at least gave some insights. We will be happy to answer any other questions. We look forward to your response.
> > >
> > > Thank you again
> > >
> > > Best
> > >
> > > All authors

---

### Official Review · Reviewer_kHKG · 2024-07-27
**Important problem, small dataset**

**Rating:** 8
**Confidence:** 4
**Correctness:** The paper seems correct overall.
**Clarity:** The paper is clearly written.

**Review:**

The experimental setup is interesting and the evaluation is relevant. There are two main weaknesses of the proposed dataset, however.

1) The dataset is small. Coming from only 22 tiles, it does not capture sufficient variability in terms of technical and biological variation. The mostly 10k patches are highly correlated because they come from the same experiment. It is unclear if the tiles come from different wells or different plates. If all come from the same experiment and there is no variation, the problem is not representative of the needs in real biological experiments.

2) It seems like the dataset will not be truly open. Researchers interested need to apply to get the dataset, which actually limits progress and future research.

**Strengths:**

Good experimental evaluation, interesting problem.

**Additional Feedback:**

See above.

**Documentation:**

Not reviewed.

**Ethics:**

No concerns.

**Limitations:**

See above.

**Opportunities For Improvement:**

The dataset needs to be larger and more diverse.
Make the dataset truly open.

**Relation To Prior Work:**

Good presentation of previous work.

**Summary And Contributions:**

The paper presents a dataset of cell images that have been captured at low and high resolution to evaluate models that predict super resolution images. The dataset was created from a single multi-well plate experiment and 22 tiles were captured with both low and high resolution. In contrast to other datasets, the low resolution images are not downscaled versions of the high resolution, but instead are captured at low resolution. The dataset is pre-processed to generate nearly 10,000 patches of 512x512 images for methods evaluation. The paper also presents an evaluation of 15 methods for super-resolution.

---

> ### Author Rebuttal · Authors · 2024-08-17
>
> Thank you for your constructive comments. We now address your comments under "Review" and "Opportunities For Improvement":
>
> - **Question: "The dataset is small and does not capture sufficient variability."**
>
> Reply:
>
> We would like to clarify the size and variability contained within the dataset. To automate the image acquisition, we programmed the microscope to capture multiple sets of 100 images. Each set of 100 images was stitched together (10x10) to form an image tile. Therefore, each tile represents 100 unique images (fields of view) and the 22 tiles correspond to 2200 unique images. Each image position was captured at 4 different resolutions (1024x1024, 512x512, 256x256, and 128x128) resulting in a dataset containing 8800 images (of 2200 unique fields of view). In addition, each of the 8800 images is a composite of 3-4 images of different molecular markers, each captured as a different channel. Collectively, this represents approximately 32,000 image elements in the dataset. We consider this a diverse and valuable dataset that will provide researchers flexibility in their use of the dataset containing multiple image parameters (4 different resolutions and 3-4 channels).
>
> For our study, we chose to split the image tiles into sampled patches ($\sim$10k) for input to SR models. We consider that this relatively large set allows training large deep learning models. Compared to the very recently published SIM datasets [32], they have only $\sim$500 patches per cell type. We believe that our dataset has a good ratio of patches per cell-type to conduct training and evaluation of deep SISR models. Note that the statistics reported in Tab.1 are per-cell type. This means that each cell-type has its own set of $\sim$10k patches per scale. Moreover, we provide 4 parallel scales at once per patch making our dataset extremely rare and more valuable for research since datasets employed in the literature have very few scales such as in [32].
>
> Note that it is challenging to quantify diversity in this context. Also, the notion of diversity could change depending on the task at hand.
>
> To address the question about diversity contained within the dataset, we performed an object-based analysis of cellular structures captured in the image dataset. The 2200 unique image fields of view contain $\sim$16,800 multi-cellular objects. The cells were cultivated in a three-dimensional protein matrix that allows them to form tissue-like cyst structures that resemble the natural organization of cells in tissues and organs, such as the colon, lung, breast, etc. Cells grown in a three-dimensional matrix exhibit spontaneous variations in cyst size and organization (hollow or solid). We recently [REF1, REF2] reported multiple cellular phenotypes in Caco2 cells that resemble tissue geometries found in healthy and disease states. Our dataset contains cysts with size variations over 2 orders of magnitude. Moreover, shape analysis of the cysts demonstrates substantial variation in aspect ratio and circularity. This reflects cysts covering a spectrum of phenotypes, including single-layered, multi-layered, and solid.  This contrasts with cells grown as two-dimensional cultures on plastic or glass surfaces, where there is substantially less variation in size and organization when viewing single cells or confluent patches as is often used in other recently published microscopy datasets. You find the analysis of the cells in our data in this figure: [link](https://drive.google.com/file/d/1QOEBwW7iMtTSErwcYfvAS9zMZ3FhUopn/view?usp=sharing). We can observe from these plots that the cells span a large size range with a relatively circular shape.
>
> There is also diversity in the molecular markers included in the images, which have different subcellular positions and expression levels. Therefore, the dataset contains substantial cellular variation that represents a wide range of cellular phenotypes.
>
> Our dataset also includes diverse cellular markers with unique properties. Cells were labeled with Hoechst dye that labels chromatin and is frequently used as a nuclear stain in cell biology. We also include mCherry-Histone, a red-fluorescent protein conjugated to histone H2B that is part of chromatin. This serves as an additional marker of cell nuclei and represents a signal that is often employed in live imaging experiments. mCherry-Histone represents a lower signal than Hoechst because the cells were selected to have a low-to-moderate expression of this protein and are detected by direct fluorescence. The other two markers (E-cadherin and Survivin) represent typical staining obtained with indirect fluorescence methods, using primary and fluorophore-conjugated secondary antibodies. E-cadherin is expressed in all Caco2 cells and is a cell surface protein. Therefore, this has a distinct expression pattern from the blob-like nuclear labels described above. Despite being expressed in all cells, there is heterogeneity at the cellular level, providing extensive natural variation within our dataset. Survivin is transiently expressed in cells during cell division. This represents an additional unique marker pattern that appears as an intense patch connecting two cells. Survivin is only expressed in a subset of cells during mitosis and cytokinesis, processes that often utilize imaging and machine learning for analysis [REF3].
>
> **Important: due to the limited space, we provide more answers to the rest of your questions in the general rebuttal toward the end: [here](https://openreview.net/forum?id=GtYd9PCaaB&noteId=7pZ2UcuHw3). Please consider them. We apologize for the inconvenience.**
>
> In our revised manuscript, we will clarify the points described above and in the additional response. We thank you again for your helpful review and hope that our response will justify raising your overall rating of our manuscript.

---

> > ### Author Response · Authors · 2024-08-21
> > **follow up**
> >
> > Dear Reviewer "kHKG",
> >
> > We sincerely thank you for your precious time and efforts in reviewing our paper. We believe our responses have addressed your concerns including the dataset size/collection/variability, and its accessibility. If you have any additional questions or require further clarification, we would be happy to provide it.
> >
> > We are looking forward to your reply.
> >
> > Best -- The Authors

---

> > ### Comment · Reviewer_kHKG · 2024-08-27
> >
> > Thank you for clarifying and explaining the phenotypic diversity of the data. That was very helpful, and the dataset will be a great resource for further research.
> > I still don't understand why the dataset needs to be restricted, and this was not justified in the rebuttal. I am willing to increase the score more if the dataset is made publicly available with a CC license.

---

> > > ### Author Response · Authors · 2024-08-29
> > > **Dataset license**
> > >
> > > Dear Reviewer "kHKG",
> > >
> > > We are glad that you found our answer to the dataset diversity aspect satisfactory, and that you increased your score.
> > > We just want to note that we have mentioned our answer to the question about sharing the data in the rebuttal ([here](https://openreview.net/forum?id=GtYd9PCaaB&noteId=7pZ2UcuHw3)). At the submission, the dataset is accessible upon request following common standards.
> > >
> > > However, following your advise, and after consulting our collaborator from the biology team, all parties have agreed to make the dataset available under Creative Commons Attribution-NonCommercial-ShareAlike 4.0 International license (CC BY-NC-SA 4.0). The download link is now public and users do not need to do a request.
> > > These changes have already been made on the dataset website (https://github.com/sbelharbi/sr-caco-2?tab=readme-ov-file#download-ds), and they will be also included in the paper.
> > >
> > > We hope that this will help further increase your score.
> > > We thank you again for your valuable feedback.
> > >
> > > Best
> > >
> > > The authors

---

> > > > ### Author Response · Authors · 2024-09-01
> > > > **reminder**
> > > >
> > > > Dear Reviewer "kHKG",
> > > >
> > > > We thank you again for your effort to review our work. This is a kind reminder that the deadline for discussion is closing in less than **1 hour 50min**. We hope that you will be able to see our response regarding the license. We followed your recommendation and used a CC license.  We hope that this will help further increase your score. We thank you again for your valuable feedback. We look forward to your response.
> > > >
> > > > Thank you again
> > > >
> > > > Best
> > > >
> > > > All authors

---

> > > > > ### Comment · Reviewer_kHKG · 2024-09-04
> > > > >
> > > > > Thank you for considering this request and for making the dataset publicly available with a CC license. This will be a great resource for the community!

---

> > > > > > ### Author Response · Authors · 2024-09-05
> > > > > > **response**
> > > > > >
> > > > > > Dear Reviewer "kHKG",
> > > > > >
> > > > > > Thank you for getting back to this, considering our response, and increasing your score. We are glad our response was satisfactory.
> > > > > >
> > > > > > Thank you again
> > > > > >
> > > > > > Best
> > > > > >
> > > > > > All authors

---

### Author Rebuttal · Authors · 2024-08-17

We thank the reviewers for their constructive feedback which is helping us improve the quality of our manuscript. We are encouraged by their positive comments, in particular:

- "Good experimental evaluation, interesting problem." (**kHKG**).
- "This is an exciting and beautiful work. Create a large-scale dataset. Benchmarking 15 methods, which shows a clear picture of how state-of-art methods perform. Give clear instructions for the public to access the dataset." (**wsnJ**)
- "The paper is recommended for acceptance. This paper is well-written, clearly demonstrating the significance of the proposed dataset. The experimental results of the benchmark deep learning models are also convincing." (**zxUt**)


We have carefully considered the reviewers’ comments. In the following, we will briefly clarify some general points about the purpose of the paper, the supplementary material, review guidelines, and the dataset.

- This work mainly proposes a new dataset for confocal microscopy super-resolution. Our additional effort to benchmark 15 representative state-of-the-art SISR methods is a secondary contribution. The comprehensive analysis of their performance and limitations can provide a reference for future work on SISR with our new SR-CACO-2 dataset. (Reviewer **zpaL**)
- This submission comes with supplementary materials as required by track guidelines. We kindly invite the reviewers to consult our supplementary materials since they provide many additional details related to the SR-CACO-2 dataset (e.g., its limitations) and the benchmarking experiments. (Reviewer **zxUt**)
- The review in NeurIPS Datasets and Benchmarks Track is primarily single-blind. (Reviewer **zpaL**)
- The $\sim$10k patches presented in Tab.1 in the main paper are statistics per single cell, and for a single scale. Our dataset is comprised of patches from 3 cell types and 4 scales. (All reviewers)


We thank all the reviewers again for their constructive comments.
These comments will be addressed to improve the camera-ready version of our paper. Please refer to our detailed responses to individual reviewer comments. Please see our reviewer-specific feedback for more information.

Below are new references used in the rebuttal with all reviewers:

- [REF1]: Wang LT, Rajah A, Brown CM, McCaffrey L. CD13 orients the apical-basal polarity axis necessary for lumen formation. Nature Communications. 2021 Aug 4;12(1):4697.[Link](https://pubmed.ncbi.nlm.nih.gov/34349123/).
- [REF2]: Wang LT, Proulx ME, Kim AD, Lelarge V, McCaffrey L. A proximity proteomics screen in three-dimensional spheroid cultures identifies novel regulators of lumen formation. Scientific reports. 2021 Nov 23;11(1):22807. [Link](https://pubmed.ncbi.nlm.nih.gov/34815476/).
- [REF3]: Khushi M, Dean IM, Teber ET, Chircop M, Arthur JW, Flores-Rodriguez N. Automated classification and characterization of the mitotic spindle following knockdown of a mitosis-related protein. BMC bioinformatics. 2017 Dec;18:149-59. [Link](https://pubmed.ncbi.nlm.nih.gov/29297284/).
- [REF4]: Cai J, Zeng H, Yong H, Cao Z, Zhang L. Toward real-world single image super-resolution: A new benchmark and a new model. ICCV, 2019.

Note that references under the form [3, 4, 5] and so on, are meant to indicate references from the submitted paper. However references with the form [REFx] are new and fully mentioned here in this general comment.

**More rebuttal for reviewer [kHKG](https://openreview.net/forum?id=GtYd9PCaaB&noteId=JewSTnJfwJ):**

Reviewer **kHKG** has raise important questions. Due to limited space, we use this section to continue our response to their questions. We apologize for the inconvenience:

- **Question: "Do tiles come from different wells/plates?"**:

Reply:

Each tile of 100 images was captured in a multi-well plate.  To simplify data manipulation, we stitched the 10x10 grid back into a single large image. The 22 wells were collected from 4 independent experiments using 5-6 wells per plate/experiment. This is considered adequate biological diversity. We will certainly clarify this better in the paper.

As described above, the images were collected from three-dimensional cultures. To establish these, single-cell suspensions were seeded into multi-well dishes in a protein matrix. The cells were grown for up to 2 weeks to form cysts. The cells were then processed using the standard 2-day method for immunostaining cells for the indicated markers (fixation in paraformaldehyde, cell permeabilization, blocking non-specific binding, incubation in marker-specific primary antibodies, incubation in fluorescently labeled secondary antibodies, staining with Hoechst). The microscope was then programmed to collect tiles of 100 images from the 4 different resolutions and would then perform image acquisition over $\sim$12-16 hours. This would result in the capture of up to 6 sets of tiles at different resolutions, so it is relatively time-consuming to generate the dataset.

- **Question: "The dataset is not truly open."**

Reply:

Accessing a dataset through a request is a very common procedure in different machine learning fields. We note that we have received several requests to access the data and we have fulfilled our promise by providing the users access without issue. However, if this aspect is contributing to your low score, we can waive the request, allowing users to directly download the dataset without sending us a request.

---

### Decision · Program_Chairs · 2024-09-26

**Decision:**

Accept (Poster)

**Comment:**

This submission introduces the SR-CACO-2 dataset for confocal microscopy super-resolution, alongside a benchmark of 15 state-of-the-art methods. Three reviewers responded positively, praising the dataset's significance, novelty, and the detailed benchmarking. They also appreciated the clear documentation and public accessibility of the dataset, noting its relevance to addressing real-world super-resolution challenges in microscopy. The deep learning model evaluations were seen as robust, offering insights for further research.
One reviewer raised concerns about the dataset’s size and diversity, questioning its adequacy for capturing real-world biological variability. The authors responded thoroughly, clarifying the dataset’s biological diversity and making it freely available under a Creative Commons license, which addressed the reviewer’s concerns and led to an improved score.